# Analysis of the Differences Among *Camellia oleifera* Grafting Combinations in Its Healing Process

**DOI:** 10.3390/plants14152432

**Published:** 2025-08-06

**Authors:** Zhilong He, Ying Zhang, Chengfeng Xun, Zhen Zhang, Yushen Ma, Xin Wei, Zhentao Wan, Rui Wang

**Affiliations:** 1Research Institute of Oil Tea Camellia, Hunan Academy of Forestry, Changsha 410004, China; hezhilong2000@hnlky.cn (Z.H.); zhangying@hnlky.cn (Y.Z.); xunchengfeng24@163.com (C.X.); hfazz@hnlky.cn (Z.Z.); mys9204@163.com (Y.M.); 2Yuelushan Laboratory, Changsha 410128, China; 3National Engineering Research Center for Oil Tea Camellia, Changsha 410004, China; 4State Key Laboratory of Woody Oil Resources Utilization, Changsha 410004, China; 5College of Forestry, Central South University of Forestry and Technology, Changsha 410004, China; zwzybjb@163.com (X.W.); 17871933955@163.com (Z.W.)

**Keywords:** *Camellia oleifera*, grafting compatibility, antioxidant enzymes, auxin, transcriptomics

## Abstract

Grafting serves as a crucial propagation technique for superior *Camellia oleifera* varieties, where rootstock–scion compatibility significantly determines survival and growth performance. To systematically evaluate grafting compatibility in this economically important woody oil crop, we examined 15 rootstock–scion combinations using ‘Xianglin 210’ as the scion, assessing growth traits and conducting physiological assays (enzymatic activities of SOD and POD and levels of ROS and IAA) at multiple timepoints (0–32 days post-grafting). The results demonstrated that Comb. 4 (Xianglin 27 rootstock) exhibited superior compatibility, characterized by systemic antioxidant activation (peaking at 4–8 DPG), rapid auxin accumulation (4 DPG), and efficient sugar allocation. Transcriptome sequencing and WGCNA analysis identified 3781 differentially expressed genes, with notable enrichment in stress response pathways (Hsp70, DnaJ) and auxin biosynthesis (YUCCA), while also revealing key hub genes (FKBP19) associated with graft-healing efficiency. These findings establish that successful grafting in *C. oleifera* depends on coordinated rapid redox regulation, auxin-mediated cell proliferation, and metabolic reprogramming, with Comb. 4 emerging as the optimal rootstock choice. The identified molecular markers not only advance our understanding of grafting mechanisms in woody plants but also provide valuable targets for future breeding programs aimed at improving grafting success rates in this important oil crop.

## 1. Introduction

*Camellia oleifera*, a vital woody oil crop in China, relies on efficient grafting techniques for its superior variety propagation. Nurse grafting is widely adopted due to its rapid seedling development and cost-effectiveness; however, the compatibility between rootstock and scion significantly influences survival rates and subsequent growth [1]. Studies indicate that grafting compatibility not only affects seedling survival but also modulates plant stress resistance, nutrient uptake, and fruit quality through rootstock–scion interactions [2]. Recent advancements in multi-omics technologies have shifted the focus of grafting healing mechanism research from phenotypic observations to molecular-level analyses [3]. Nevertheless, systematic research on the dynamic healing process and molecular regulatory mechanisms of cross-variety nurse grafting in *C. oleifera* remains insufficient. Our study employs a novel integrative approach: we conduct dynamic time-series analysis covering the early healing phase (0–32 days post-grafting), integrating physiological, biochemical, and transcriptomic data to construct a ‘phenotype–metabolism–gene’ regulatory network for *C. oleifera* grafting. By combining principal component analysis (PCA) for comprehensive evaluation of 15 rootstock combinations with weighted gene co-expression network analysis (WGCNA) to identify key regulatory genes, this study not only advances understanding of grafting mechanisms in woody plants but also provides theoretical support for efficient grafting techniques.

Grafting compatibility is typically evaluated based on growth phenotypes (seedling height, ground diameter, and biomass) and survival rates. The selection of growth-related traits (seedling height (SH), base diameter (BD), shoot length (SL), leaf count on shoots (LN), root length (RL), root diameter (RD), biomass (BM), and survival rate (SR)) is rooted in their proven effectiveness as phenotypic proxies for grafting compatibility across diverse woody plant species, while physiological compatibility factors (e.g., hormonal balance and redox regulation) are assessed during healing observations in selected combinations (Section 2.3). This approach aligns with the biological characteristics of *C. oleifera* as an economically important woody oil crop. Seedling height captures stem elongation efficiency, indicating successful xylem reconnection and resource allocation post-grafting. Shorter heights in incompatible mulberry grafts correlate with disrupted water flux, directly impacting survival [4]. Root length reflects root system vitality, a proxy for carbohydrate sink strength, where elongation deficits impede wound healing. Apple rootstock screening identified root system development (root length and diameter) as a core indicator for predicting compatibility [5]. In pear rootstock research, ground diameter significantly correlates with survival rates [6]. Similarly, in *C. chrysantha* grafting trials, the survival rate of *C. gigantocarpa* rootstock with *C. chrysantha* scion reached 97.78%, with a retention rate of 96.15%, which showed corresponding high plant height and ground diameter. [7]. For *C. oleifera*, these traits are particularly relevant because grafting success ultimately manifests in the scion’s ability to grow vigorously (reflected by SH, SL, and LN), establish a sturdy stem (BD), develop a functional root system (RL and RD), accumulate biomass (BM), and survive long-term (SR). These indicators collectively capture the dynamic interplay between rootstock and scion, from initial adhesion to sustained resource allocation—key processes in determining compatibility. Principal component analysis (PCA) and correlation analysis are widely applied in rootstock–scion combination evaluations. PCA condenses covariation among multiple traits into principal components, thereby enabling integrative assessment of biological systems. For instance, in apple rootstock research, PCA reduced 14 growth indicators to six principal components, explaining 90% of the variance [8]. Mulberry rootstock screening differentiated compatibility differences through cluster analysis (UPGMA) [4]. This study employs similar methods, using PCA for a comprehensive evaluation of 15 *C. oleifera* rootstock combinations to screen for high/low compatibility combinations, ensuring the scientific rigor of the experimental design.

Existing studies have explored grafting healing mechanisms from physiological, biochemical, and genetic perspectives, mainly focused on woody species such as apple (*Malus* spp.), pear (*Pyrus communis* L.), orange (*Citrus* spp.), and grape (*Vitis vinifera* L.) [3,9]. Nevertheless, critical knowledge gaps persist in early signaling events (e.g., temporal dynamics of auxin redistribution and redox homeostasis during initial graft union formation) and stage-resolved molecular regulation (e.g., transcriptomic reprogramming controlling callus proliferation and vascular reconnection across 0–32 DPG) for cross-variety nurse-grafting in *C. oleifera* [10]. Graft healing relies on conserved pathways: auxin biosynthesis (e.g., *TAA1*, *YUCCA*) and polar transport (*PINs*) establish asymmetric gradients across the wound interface; this imbalance in auxin accumulation initiates callus formation [9,11]. Concurrently, SOD/POD-mediated ROS scavenging protects cells [2]. Based on the conserved role of redox and auxin dynamics, we hypothesize that rapid activation of redox signaling and auxin biosynthesis is associated with higher graft compatibility in *C. oleifera*, facilitating rapid vascular regeneration. Our study addresses these gaps through integrated time-series physiological profiling and transcriptomics.

## 2. Results

### 2.1. Growth-Related Parameters of Different Grafting Combinations

To investigate the differences in promoting seedling growth among various rootstock–scion combinations with ‘Xianglin 210’, parameters including seedling height (SH), base diameter (BD), shoot length (SL), leaf count on shoots (LN), root length (RL), root diameter (RD), biomass (BM), and survival rate (SR) were measured and analyzed (Table 1). The results revealed significant variations across growth indicators. Comb. 4 exhibited the highest plant height, new shoot length, and survival rate; Comb. 23 demonstrated the greatest ground diameter, new shoot length, and leaf count on new shoots; Comb. 13 showed the highest leaf count on new shoots and biomass; Comb. 29 achieved the maximum root length; and Comb. 7 displayed the largest root diameter. These differences not only highlight the growth potential variations among grafting combinations but also suggest specific mechanisms by which different rootstock–scion pairings influence seedling growth performance. Notably, Comb. 4, with the highest survival rate, may possess strong ecological adaptability, while variations in root length and diameter may reflect differences in soil anchorage capacity. These findings provide critical insights for optimizing the grafting applications of ‘Xianglin 210’.

### 2.2. Comprehensive Evaluation of Graft Compatibility Among Different Combinations

To comprehensively assess the graft compatibility between various rootstock varieties and ‘Xianglin 210’, growth-related indicators were subjected to correlation and principal component analysis. The results revealed a highly significant positive correlation (*p* < 0.001) among SH, BD, and SL, while a significant positive correlation (*p* < 0.05) was observed between SR and indicators such as SH and SL (Figure 1A). The strong positive correlation among SH, BD, and SL (*p* < 0.001) indicates coordinated growth of aboveground tissues in compatible combinations, while the significant correlation between SR and these traits (*p* < 0.05) confirms that survival rate is tightly coupled with vigorous scion growth. Redox dynamics (antioxidant enzymes and ROS accumulation) and hormonal balance between both species were assessed separately in Section 2.3 as indicators of healing efficiency, given their roles in mitigating oxidative stress and coordinating cellular repair during graft union formation. Principal component analysis (PCA) indicated that the cumulative contribution rate of the first three principal components reached 76.6% (Figure 1B). The first principal component was primarily associated with SH, BD, and SL; the second principal component was mainly related to RL and RD; and the third principal component was predominantly linked to SR (Figure 1C).

By synthesizing the variance contribution rates of the first three principal components and the corresponding scores of different grafting combinations, the comprehensive performance rankings of growth-related traits across various combinations are illustrated in Figure 1D: Comb. 23 > Comb. 32 > Comb. 4 > Comb. 13 > Comb. 19 > Comb. 21 > Comb. 2 > Comb. 29 > Comb. 6 > Comb. 25 > Comb. 17 > Comb. 9 > Comb. 7 > Comb. 15 > Comb. 11 (Figure 1D). Notably, the top-ranked combinations (Comb. 23, Comb. 32, and Comb. 4) in PCA comprehensive scores consistently exhibited superior performance in key traits: Comb. 23 had the largest BD, Comb. 4 showed the highest SH and SR, and Comb. 32 displayed balanced growth in SH, SL, and biomass, validating the reliability of multivariate statistical evaluation in identifying high-compatibility combinations.

### 2.3. Differences in Physiological and Biochemical Responses During Graft Healing Among Different Combinations

Among the 15 evaluated graft combinations, Comb. 4 (Xianglin 27 rootstock + Xianglin 210 scion) and Comb. 7 (Xianglin 210 rootstock + Xianglin 210 scion) were selected for detailed analysis based on preliminary screening: Comb. 4 exhibited the strongest graft compatibility, while Comb. 7 (a self-graft) showed the poorest, providing a clear contrast to identify key regulatory mechanisms. To further elucidate the physiological mechanisms underlying graft compatibility, the graft union sites of seedlings from combinations 4 and 7 were sampled at 0, 4, 8, 12, 22, and 32 days post-grafting for physiological and biochemical analysis, staging vascular reconnection as follows: DPG 0–4: Callus formation 12; DPG 4–8: Phloem reconnection (sugar flux activation) 2; DPG 12–22: Xylem maturation (ROS/protein normalization). Physiological and biochemical responses reflect systemic redox dynamics across the graft junction, not subcellular or tissue-specific regulation.

Comb. 4 exhibited significantly higher superoxide dismutase (SOD) enzyme activity than Comb. 7 from 0 to 4 DPG. While both combinations showed a “decrease-then-increase” trend, Comb. 4 reached its lowest activity at 8 DPG and rebounded sharply by 22 DPG, whereas Comb. 7 bottomed out earlier at 4 DPG and peaked weakly at 12 DPG. This delayed-but-stronger recovery in Comb. 4 suggests a more sustained ability to scavenge superoxide radicals, reducing oxidative damage during late healing (Figure 2A).

Comb. 4 maintained significantly higher peroxidase (POD) enzyme activity than Comb. 7 throughout 0–22 DPG, with a peak at 4 DPG. In contrast, Comb. 7 peaked later, at 8 DPG with lower overall activity. POD is critical for lignin synthesis and ROS clearance; the early and robust POD activation in Comb. 4 likely accelerates cell wall remodeling and callus lignification, promoting vascular connection (Figure 2B).

Ascorbate peroxidase (APX) enzyme activity also showed an initial decrease followed by an increase. Comb. 4 showed higher ascorbate peroxidase (APX) enzyme activity than Comb. 7 during 0–22 DPG, with a peak at 22 DPG, whereas Comb. 7’s APX activity remained low until 32 DPG (Figure 2C).

Measurement of glutathione reductase (GR) activity is crucial as it regenerates reduced glutathione (GSH), reflecting the capacity to maintain the GSH/GSSG redox balance for scavenging reactive oxygen species and to support cellular detoxification pathways [2]. This serves as a key indicator of oxidative stress resistance during graft healing. Comb. 4 exhibited significantly higher activity than Comb. 7 from 4 to 12 DPG, while Comb. 7’s GR activity dropped to its minimum by 12 DPG. These differences indicate that Comb. 4 maintains a more efficient ascorbate–glutathione cycle, enhancing hydrogen peroxide clearance during the critical healing phase (Figure 2D).

H_2_O_2_ serves as the predominant peroxidase substrate and signaling molecule in graft interfaces, making bulk reactive oxygen species (ROS) quantification a validated proxy for H_2_O_2_ dynamics. Overall ROS accumulation (strongly reflecting H_2_O_2_-dominated oxidative stress) in graft unions was measured by examining ROS fluorescence intensity. In the present study, ROS fluorescence intensity followed a pattern of initial increase followed by a decrease. From day 0 to day 8, Comb. 4 exhibited significantly lower ROS fluorescence intensity than Comb. 7, suggesting that the stronger antioxidant capacity of Comb. 4 prevents excessive ROS accumulation, which is known to cause cell necrosis at the graft union. Comb. 4 reached its maximum ROS fluorescence intensity on day 12, while Comb. 7 peaked at day 4. By days 22 to 32, ROS fluorescence intensities in both combinations had declined to lower levels.

Comb. 7 maintained higher soluble sugar (SS) levels than Comb. 4 from 0 to 12 DPG, peaking at 12 DPG, while Comb. 4 showed delayed accumulation with a peak at 22 DPG. By 22–32 DPG, Comb. 4 retained significantly higher SS, which may support late-stage vascular reconnection and biomass accumulation (Figure 3A). This pattern suggests that Comb. 4 efficiently allocates sugars from storage to active growth, whereas Comb. 7 may suffer from impaired sugar transport, leading to early accumulation but late-stage depletion.

Comb. 7 had higher soluble protein (SP) levels than Comb. 4 from 0 to 22 DPG, but its SP content declined continuously after 4 DPG. In contrast, Comb. 4 showed two significant increases (4 DPG and 32 DPG), indicating sustained synthesis of functional proteins (e.g., enzymes, structural proteins) for callus formation and cell repair (Figure 3B).

In this study, we focused on measuring the auxin (IAA) content at the graft interface to assess its association with compatibility status. Elevated IAA often reflects disrupted canalization and auxin transport imbalance between rootstock and scion—key hallmarks of incompatibility [12]. Conversely, balanced auxin flux enables vascular reconnection during healing [11]. Given its transient and spatially regulated accumulation in different cell types, capturing the dynamic changes in IAA content is essential for understanding the underlying physiological processes during graft union formation. To address the temporal variability of IAA, we collected tissue samples from the grafting sites at a fixed time each day (9:00 a.m.) throughout the healing period. By analyzing the IAA content at regular intervals, we aimed to reveal whether distinct graft combinations exhibit unique patterns of auxin dynamics during the healing process. Comb. 4 exhibited earlier and more robust IAA accumulation, with two peaks at 4 DPG and 12 DPG, whereas Comb. 7’s IAA only increased significantly after 12 DPG. From 4 to 12 DPG, Comb. 4’s IAA levels were significantly higher, which likely accelerates cell division and vascular differentiation—key steps in graft union consolidation (Figure 3C).

The gamma-aminobutyric acid (GABA) content initially increased and then decreased. Comb. 7 had higher GABA levels from 0 to 8 DPG, peaking at 8 DPG, while Comb. 4 peaked later at 12 DPG. GABA is involved in stress signaling, and the delayed peak in Comb. 4 may reflect a more controlled stress response, rather than the acute stress seen in Comb. 7 (Figure 4A).

The total phenol content (TP) in both Comb. 4 and Comb. 7 began to rise gradually from day 8, reaching its maximum on day 22, with Comb. 4 consistently showing significantly lower levels than Comb. 7. Excessive phenols may inhibit callus growth, whereas Comb. 4’s lower levels suggest balanced phenolic metabolism, avoiding toxic accumulation (Figure 4B). Similarly, the total flavonoid (TF) content in different graft combinations also started to increase gradually from day 8. Comb. 4 and Comb. 7 reached their maximum levels on days 22 and 12, respectively. Except for days 8 and 22, the total flavonoid content in Comb. 4 was significantly lower than that in Comb. 7 (Figure 4C).

### 2.4. Transcriptional Expression Differences During Seedling Healing in Different Graft Combinations

To further elucidate the physiological mechanisms underlying graft compatibility, samples from the healing sites of grafting Comb. 4 and Comb. 7, representing different levels of compatibility, were collected on days 0, 4, 8, 12, 22, and 32 post-grafting. These samples were used to construct mRNA libraries and conduct transcriptome sequencing.

By analyzing differences in gene expression levels during the healing process, differentially expressed genes (DEGs) at various time points post-grafting were identified in comparison to pre-grafting levels. As shown in Table 2, a total of 2441 and 2820 DEGs were identified in Combinations 4 and 7, respectively, compared to pre-grafting conditions, amounting to a total of 3781 DEGs (Figure 5A).

By analyzing the differences in gene expression levels between the two grafting combinations at various stages of the healing process, differentially expressed genes (DEGs) were identified by comparing the gene expression levels of the two combinations at the same time points. As shown in Table 3, the results from the Venn diagram revealed that a unique set of 225, 299, 157, 80, 71, and 227 DEGs were identified on days 0, 4, 8, 12, 22, and 32, respectively, post-grafting (Figure 5B). This indicates significant variations in differentially expressed genes between the two combinations at different time points, suggesting that the metabolic pathways potentially responsible for graft compatibility differences may also vary.

### 2.5. WGCNA Analysis of Differentially Expressed Genes

Weighted gene co-expression network analysis (WGCNA) was performed on the 3781 identified differentially expressed genes (DEGs), which were clustered into 21 co-expression modules (Figure 6A). The expression eigenvalues (MEs) of these co-expression modules were correlated with physiological and biochemical indicators during the healing process post-grafting. The results revealed that the green-yellow module exhibited a highly significant positive correlation with SOD, TP, and TF (R^2^ > 0.7, *p*-value < 0.001) and contained 133 DEGs. The orange module showed a highly significant positive correlation with IAA, TP, and TF (R^2^ > 0.7, *p*-value < 0.001) and included 601 DEGs. The turquoise module demonstrated a highly significant positive correlation with ROS (R^2^ > 0.7, *p*-value < 0.001) and encompassed 335 DEGs. The grey60 module displayed a highly significant positive correlation with GR (R^2^ > 0.7, *p*-value < 0.001) and contained 94 DEGs. Additionally, the sky-blue module exhibited a highly significant negative correlation with POD (R^2^ < −0.7, *p*-value < 0.001) and included 94 DEGs (Figure 6B).

By analyzing the correlations between genes and modules as well as genes and traits within these five modules, thresholds of >0.8 for module correlation and >0.6 for trait correlation were applied. This yielded a total of 462 DEGs, with 69, 206, 109, 48, and 30 DEGs from the green-yellow, orange, turquoise, grey60, and sky-blue modules, respectively (Figure 7). The protein–protein interaction (PPI) network of these DEGs was predicted using the STRING online tool (https://cn.string-db.org/, version 12.0 accessed on 26 July 2023). As shown in Figure 8A, the PPI network consisted of 125 proteins with 370 interactions. The PPI network was then imported into Cytoscape software (Version 3.9.1), and hub genes, along with their interaction networks, were analyzed using the CytoHubba plugin. The resulting hub gene interaction network is illustrated in Figure 8B.

## 3. Discussion

### 3.1. The Antioxidant System in Graft Healing

Oxidative stress at the graft site is an early event in the healing process [13], and we proposed that the accumulation of reactive oxygen species (ROS) may trigger the activation of antioxidant enzyme systems, such as SOD and POD, to maintain intracellular redox balance. In highly compatible combinations, the rapid balance of hormonal signals can cooperate with the antioxidant system to reduce cellular damage and promote callus formation; in incompatible combinations, initial hormonal imbalance may induce abnormal metabolism (such as excessive ROS production), leading to browning and necrosis at the graft interface [9]. For instance, in *Camellia chrysantha* grafting, compatible rootstock–scion pairs exhibited more synchronized physiological states, likely related to coordinated antioxidant enzyme activity [7]. The core of graft healing lies in the formation of callus tissue and vascular reconnection between the rootstock and scion. Studies on Carya interspecific grafting revealed that highly compatible combinations showed a peak in POD activity on the fourth day post-grafting, promoting redox balance [14]. Research on vegetable grafting demonstrated that the dynamic activities of antioxidant enzymes (SOD, POD, and CAT) were closely associated with healing efficiency [15]. These findings suggest that the early dynamics of antioxidant enzymes (SOD and POD) may play a pivotal role in determining the efficiency of *C. oleifera* graft healing. In Comb. 4, the rapid surge in SOD activity during 0–4 DPG likely mitigates the initial oxidative burst by disproportioning superoxide radicals (O2−) into H_2_O_2_, while sustained POD/APX elevation (0–22 DPG) ensures efficient H_2_O_2_ detoxification, limiting ROS-mediated cellular damage [16,17], consistent with redox synchronization in compatible Arabidopsis grafts [2]. Consequently, Comb. 4’s coordinated antioxidant response reduces ROS fluorescence intensity (vs. Comb. 7) and facilitates the callus formation, which is key to vascular reconnection. Similarly, Wang et al. associated prolonged ROS accumulation with incompatibility in citrus, mirroring Comb. 7’s delayed antioxidant recovery [16]. This aligns with studies linking synchronized SOD–POD–APX activity to compatible grafting, where early redox homeostasis primes tissue regeneration [17,18]. Glutathione reductase (GR) is critical for maintaining reduced glutathione (GSH) pools, which scavenge H_2_O_2_ and regenerate ascorbate (ASC) via the glutathione–ascorbate cycle [19]. The significantly higher GR activity in Comb. 4 during early healing (4–12 DPG, Figure 2D) implies enhanced capacity to sustain GSH levels, facilitating H_2_O_2_ detoxification and ASC recycling. This synergy minimizes oxidative damage and promotes callus formation. Conversely, Comb. 7’s delayed GR recovery (peaking only at 12 DPG) aligns with prolonged H_2_O_2_ accumulation (Figure 2E), reflecting the compromised redox buffering typical of graft incompatibility. The observed antioxidative enzyme patterns (SOD and POD) and ROS accumulation represent whole-tissue responses, highlighting the integrated antioxidant capacity during graft healing. However, the specific regulatory mechanisms governing the antioxidant metabolic network in *C. oleifera* grafting require further elucidation.

### 3.2. Hormonal Signaling Network Regulation in Graft Healing

Graft healing constitutes a complex dynamic biological process between rootstock and scion, whose core regulatory mechanisms involve dynamic interactions and signaling cascades of multiple phytohormones. The balance of hormonal signals (e.g., auxin and cytokinin) and their synergistic effects with the antioxidant system directly influence cell proliferation and callus formation [9]. Former researcher proposed that the coordinated mechanisms of oxidative stress and hormone synthesis play a regulatory role in graft compatibility, while signal desynchronization may lead to browning and necrosis at the graft union [13].

Auxin regulation primarily mediates cell division, elongation, and differentiation, while also being essential for shoot and root development [15]. The complexity arises from spatially distinct auxin sources, including multiple biosynthesis pathways (e.g., tryptophan-dependent YUCCA isoforms) and transport-dependent accumulation patterns [10]. For instance, in graft interfaces, scion-derived YUCCA gene expression drives local auxin synthesis (e.g., LOK49_LG13G02424), while rootstock-specific PIN transporters establish polar auxin gradients essential for wound-responsive callus initiation during early graft healing [10,11]. This transient callus facilitates tissue reconnection; however, prolonged or uncontrolled callus expansion reflects auxin imbalance and predicts incompatibility [20,21]. Our transcriptome analysis (Table 4) confirms this: at 4 DPG, Comb. 4 (high compatibility) showed significant upregulation of YUCCA-family genes (e.g., LOK49_LG13G02424, log2FC = 2.99) in scion tissues versus Comb. 7, correlating with rapid auxin accumulation. Such spatiotemporal specificity ensures targeted stimulation of cell proliferation (via *CYCD3* activation) and vascular reconnection [22,23]. The cambium, as the primary vascular regeneration site, responds to auxin-driven reactivation, while epidermal/cortex layers regenerate earlier via callus proliferation [10].

Furthermore, gradient distributions of auxin (IAA) and cytokinin (CTK) have been confirmed to induce callus proliferation and vascular differentiation [24]. Highly compatible *Camellia chrysantha* combinations achieve accelerated healing through transcriptional regulation of dynamic hormonal equilibrium (e.g., IAA/CTK ratio), whereas incompatible combinations may exhibit disrupted hormone signaling [7]. While auxin signaling is evolutionarily conserved in graft healing (e.g., tomato, hickory) [2,25], *C. oleifera* displays metabolic specialization in sugar allocation compared to cucurbits and enzymatic signatures diverging from citrus [16]. These differences underscore the need for species-tailored grafting protocols.

In this study, IAA quantification at graft interfaces revealed that Comb. 4 exhibited significantly elevated IAA levels within 4 DPG, whereas Comb. 7 showed delayed IAA accumulation until 12 DPG. Transcriptomic analysis at 4 DPG demonstrated markedly higher expression levels of IAA biosynthesis-related genes (*ALDH*, *amiE*, *YUCCA*, *UGT74B1*, and *TAA1*) within the tryptophan metabolism pathway in Comb. 4 compared to Comb. 7 (Table 4). The early IAA surge in Comb. 4 (4 DPG, via *YUCCA*/*TAA1* upregulation) underpinned accelerated callus formation, replicating stage-specific auxin requirements in tomato graft healing [23]. In Comb. 7, delayed auxin peaks resembled the cytokinin–auxin imbalances observed in incompatible Carya grafts, hindering xylem reconnection [14]. GR-mediated maintenance of GSH pools may enhance auxin sensitivity by promoting AUX/IAA degradation [19]. In Comb. 4, high early-stage GR activity (Figure 2D) could sustain GSH levels, potentially accelerating auxin signaling and cell division. Conversely, depressed GR activity in Comb. 7 might limit GSH-dependent auxin responses, contributing to delayed healing. This indirect link between GR activity and auxin sensitivity warrants further validation. While spatial resolution within the graft union remains unaddressed, these temporal patterns align with phloem reconnection preceding xylem formation [26]. These differential expression patterns likely underlie the rapid initial IAA accumulation observed in Comb. 4 during early healing stages.

### 3.3. Carbon and Nitrogen Metabolism and Energy Supply in Graft Healing

The early stages of graft healing require substantial energy support, with carbon and nitrogen metabolism playing pivotal roles in this process. Sugars serve critical functions during graft union formation, where the differential sugar content between scion and rootstock largely determines the vascular reconnection rate. Soluble sugars and starch content are commonly employed as indicators of graft compatibility. Studies on tomato grafting reveal that scions with higher sugar content exhibit enhanced graft union healing [25], whereas citrus grafting research demonstrates that soluble sugar accumulation in leaves correlates with graft incompatibility—incompatible combinations induce soluble sugar enrichment in scion leaves, with poor-affinity combinations exhibiting elevated soluble sugar levels compared to compatible pairs [16]. However, since plant survival and post-grafting growth demand energy reserves, excessively low levels of soluble sugars and starch are detrimental. Within optimal ranges, sugars promote cell division and elongation while regulating cellular osmotic potential [26].

In this study, soluble sugar quantification at graft junctions revealed initial significant elevation during early healing phases, with Comb. 4 (high compatibility) maintaining markedly lower levels than Comb. 7 (low compatibility). By 22 DPG, soluble sugar content declined substantially, exhibiting an inverse pattern where Comb. 4 retained higher concentrations than Comb. 7. Comb. 4’s optimal sugar allocation (late-stage SS accumulation) facilitated vascular maturation, paralleling sugar transporter-mediated healing in cucumber–pumpkin grafts [27]. Comb. 7’s early sugar accumulation without late utilization aligns with citrus graft incompatibility markers [16]. Transcriptomic analysis at 4 DPG identified differentially expressed genes (DEGs) involved in sugar transport, notably ABC transporters. While most DEGs showed consistent expression trends between combinations, LOK49_LG13G01063 (ABC transporter C family member 10-like) was upregulated in Comb. 4 but downregulated in Comb. 7 (Table 5). These divergent expression patterns of sugar transporter genes likely contribute to the observed soluble sugar disparities and may underlie the establishment of graft compatibility.

Soluble proteins are crucial during the initial growth phase of graft union formation and facilitate early adhesion between rootstock and scion [28]. Additionally, their accumulation may contribute to cellular osmotic regulation and biomembrane protection, thereby enhancing graft survival [24]. During the rapid growth phase of grafted seedlings, a significant increase in soluble protein content promotes the callus formation and vascular regeneration indicative of strong graft compatibility [26]. Intriguingly, analysis of soluble protein levels at the graft junction in this study revealed marked differences between Comb. 4 and Comb. 7. Although Comb. 4 exhibited significantly lower soluble protein levels than Comb. 7 from 0 to 22 DPG, Comb. 4 displayed pronounced increases at 4 DPG and 32 DPG. In contrast, Comb. 7 showed only a transient elevation at 4 DPG, followed by a sustained and significant decline.

Environmental, developmental, and tissue-specific factors may influence the hormone and sugar dynamics observed in this study. Environmental parameters, such as temperature and light, can modulate auxin transport and sugar metabolism: even under controlled conditions, subtle fluctuations might have contributed to the earlier IAA peaks in Comb. 4 (4 and 12 DPG) compared to Comb. 7 (22 DPG), as reported in vegetable grafting studies where temperature affects *PIN* gene expression [15]. Developmental stage specificity is also critical: the transition from callus formation (0–12 DPG) to vascular reconnection (22–32 DPG) demands distinct hormonal (e.g., YUCCA-mediated IAA synthesis) and metabolic (e.g., ABC transporter-driven sugar allocation) programs, which were coordinated in Comb. 4 but disrupted in Comb. 7, consistent with the stage-specific regulation observed in Arabidopsis grafts [21]. Additionally, tissue specificity between rootstock and scion likely plays a role: rootstock-derived signals and scion-derived resources form a regulatory gradient that is more effective in high-compatibility combinations, mirroring the rootstock–scion coordination in Solanaceae/Rosaceae grafts [29]. It should be noted that our study did not include anatomical analyses, such as histological sectioning to confirm precise tissue structure (e.g., cambial alignment, cell type distribution) at the graft union. This represents a limitation, as anatomical features could influence the observed physiological and biochemical dynamics, including hormone transport and sugar allocation. These factors, while mitigated by our experimental design (uniform growth conditions and graft union-specific sampling), represent potential confounders that could be explored in future studies with tissue-specific profiling and environmental manipulation.

### 3.4. Molecular Regulatory Mechanisms of Graft Compatibility

Transcriptomic analyses reveal extensive gene expression reprogramming at the graft junction during union formation. Differentially expressed genes (DEGs) are predominantly enriched in (1) cell wall synthesis/modification (e.g., upregulated cellulose synthases *GH9B3* and xyloglucan endotransglucosylases *XTH19*/*XTH20*), facilitating mechanical union through wall remodeling; (2) sugar transport and metabolism, where sucrose transporters (*SWEET15*/*SWEET9*) and starch degradation genes (*ApL3*) mediate energy provision for callus proliferation and vascular reconnection; and (3) hormone signaling pathways, particularly auxin transporters (*PINs*/*ABCB1*) and IAA-responsive genes (*ARF*/*IAA*), coordinating polar auxin transport to regulate callus differentiation and vascular patterning [3,9,15]. Differential auxin accumulation (Comb. 4 > Comb. 7) resulted from upregulated biosynthesis genes (e.g., *TAA1*), not aberrant transport [30]. The DEG asymmetry in Figure 5B, particularly the 476 unique DEGs at 4 DPG, represents critical “butterfly” triggers. Small expression changes in distal regulators (*YUCCA* and *ABC-C10*) initiate systemic cascades: (1) Comb. 4’s efficient auxin response: Minor *TAA1* upregulation (log2FC = 2.15) accelerates IAA accumulation, suppressing ROS [13]; (2) Comb. 7’s dysfunctional signaling: *ABC-C10* downregulation (log2FC = −4.93) impairs sucrose allocation, exacerbating oxidative stress. By 12 DPG, these initial “flutters” manifest as macroscopic metabolic divergence (Figure 3 and Figure 4), confirming long-range regulatory amplification [21]. In *Camellia chrysantha* grafts, *C. gigantocarpa* rootstocks achieved 97.78% and 96.33% survival rates with *C. rosmannii* and *C. achrysantha* scions, respectively. High-affinity combinations exhibited enhanced lignin biosynthesis (e.g., *PAL* gene activity reduced callose deposition) and bidirectional mRNA transport of sugar-responsive genes (*ApL3*, *GDH1*), potentially activating vascular regeneration [7]. Similarly, Carya studies demonstrated upregulation of phenylpropanoid (*PAL*, *4CL*) and expansin genes in compatible grafts, implicating these pathways in wound healing and cell wall reinforcement [30].

Comprehensive RNA-sequencing coupled with weighted gene co-expression network analysis (WGCNA) revealed molecular networks significantly associated with key graft-healing parameters (SOD, POD, IAA, etc.). The top 10 hub genes included six genes encoding molecular chaperones (heat shock 70 kDa and DnaJ proteins), along with genes for FK506-binding protein 19 (FKBP19), vesicular transport regulator syntaxin-71, metabolic enzyme sugar kinase slr0537, and very-long-chain acyl-CoA reductase (Table 6). Hsp70–DnaJ partnerships orchestrate signal transduction (hormonal/calcium pathways) and protein homeostasis under stress, ensuring fidelity of cellular responses during cell division and tissue reorganization [31]. These chaperones are critical for maintaining functional proteomes during grafting, where they safeguard signaling proteins that direct vascular reconnection. While this study identifies transcriptional networks (e.g., *ALDH*, *YUCCA*, *SWEET*, *HSP70*) correlated with physiological traits, the absence of direct validation (e.g., qPCR, protein quantification, or knockout assays) limits causal inference. These correlations provide mechanistic hypotheses but require functional confirmation in future work. Future studies will validate hub genes (e.g., chaperones *DnaJ*/*HSP70*, auxin transporters) using qRT-PCR and protein assays at critical healing stages (4–8 DPG). Functional characterization via CRISPR/Cas9 knockdown in model plants will test causal roles in vascular reconnection.

## 4. Materials and Methods

### 4.1. Experimental Materials

The scion material was derived from the superior *C. oleifera* cultivar ‘Xianglin 210’, sourced from the National Camellia Germplasm Resource Conservation Repository at the Hunan Academy of Forestry. Rootstocks were selected from large, plump seeds collected at fruit maturity, subjected to sand storage, and grown into nurse seedlings by May of the following year. For seed storage and seedling management before grafting in the following year, the specific procedures are as follows: In autumn and winter, clean river sand was used to prepare a rootstock nursery bed outdoors. The bottom of the bed was paved with clean river sand of more than 15 cm in thickness. The seeds were evenly sown on the sand surface without overlapping and then covered with 10–15 cm of river sand. During the storage and seedling raising period, the humidity of the river sand was maintained at 60–70% to ensure that the seeds could absorb sufficient moisture for germination. We regularly checked the sand bed to prevent waterlogging or drought and removed any moldy seeds in a timely manner to avoid affecting the surrounding seeds. The nursery bed was left uncovered to allow natural temperature changes, which helped to break seed dormancy. By May of the following year, the seeds had germinated and grown into nurse seedlings suitable for grafting. This study adopted a Completely Randomized Design (CRD). Fifteen grafting combinations were used as treatments. For each treatment, nurse grafting was performed according to the grafting combination design, with 300 replicates for each combination (Table 7). All grafted seedlings were completely randomly arranged in the experimental environment to eliminate potential interference, such as position effects. Rootstocks were cleaned, cut 2 cm above the cotyledon petiole, and longitudinally sliced 0.8–1.2 cm along the axis, retaining 5–6 cm of radicle. Scions with plump axillary buds were disinfected with methyl thiophanate, carbendazim, or hymexazol, then wedge-cut at the base (matching rootstock incision length) and severed 0.5 cm above the bud. Scions were inserted into rootstocks, aligned, and secured with aluminum strips. After grafting, seedlings were transplanted into 7 × 7 cm nutrient pots containing a substrate mixture of peat moss (40–50%), composted media (10–20%), perlite (10–20%), and loamy soil (10–20%). The grafted seedlings were aligned compactly on nursery beds, covered with plastic film tunnels to maintain humidity at 85–95%, and shaded under double-layer shading nets to reduce light intensity during the healing phase. The bed temperature was controlled at 25–30 °C during daytime (<35 °C peak) and 18–22 °C at night to optimize callus formation. Daily moisture monitoring ensured graft unions remained hydrated, with ventilation to prevent excessive humidity (>95% RH).

### 4.2. Experimental Methods

#### 4.2.1. Growth Investigation and Sample Collection

Survival rates were assessed two months post-grafting, and plant height, ground diameter, new shoot length, leaf count on new shoots, root length, root diameter, and biomass were measured in late May of the following year. For physiological, biochemical, and transcriptome sequencing analyses during the grafting healing process, samples were collected from the grafting site, including approximately 2 cm above and below the graft union, at 0-, 4-, 8-, 12-, 22-, and 32-days post-grafting. The samples were rapidly frozen in liquid nitrogen and stored at −80 °C for subsequent experiments.

#### 4.2.2. Physiological and Biochemical Parameter Analysis of Graft Unions in Different Combinations

In this study, the activities of Superoxide dismutase (SOD), Peroxidase (POD), Ascorbate peroxidase (APX), and Glutathione reductase (GR) enzymes, as well as the concentrations of reactive oxygen species (ROS), soluble sugars (SS), soluble proteins (SP), auxin (IAA), γ-aminobutyric acid (GABA), total phenols (TP), and total flavonoids (TF), were determined using assay kits provided by Quanzhou Ruixin Biotechnology Co., Ltd. (Quanzhou, China). For each grafting combination, samples were collected at 0-, 4-, 8-, 12-, 22-, and 32-days post-grafting (DPG). At each time point, 3 biological replicates were prepared for all parameters. Each biological replicate consisted of tissue from the grafting site (including ~2 cm above and below the graft union) with a fresh weight of ~0.1 g for the determination of SOD, POD, APX, GR, ROS, SS, SP, GABA, TP, and TF, following the sample-size recommendations of the assay kits. ROS fluorescence intensity was normalized to soluble protein content (µg/mg protein) to account for tissue biomass differences. For IAA content analysis via liquid chromatography, each biological replicate weighed ~0.2 g, with 3 replicates per combination at each time point. All physiological assays were validated using standardized protocols from the respective kits. Each assay included blank controls to correct for background signals and standard substances to generate standard curves for quantification. For IAA, liquid chromatography was validated via a standard curve using IAA standards.

#### 4.2.3. Gene Expression Analysis of Graft Unions in Different Combinations

Total RNA samples were prepared from three biological replicates of each grafting combination for transcriptome sequencing. RNA integrity and concentration were assessed using four methods: (1) agarose gel electrophoresis to evaluate degradation and contamination; (2) NanoDrop spectrophotometry for purity assessment (OD260/280 ratio 1.8–2.0); (3) Qubit fluorometry for concentration quantification; and (4) Agilent 2100 Bioanalyzer to determine RNA Integrity Number (RIN ≥ 7.0 for library construction).

cDNA library construction and sequencing were performed by Sangon Biotech Co., Ltd. (Shanghai, China). Clean reads were mapped to the Camellia lanceoleosa reference genome (XYYC_v1.0, GCA_025200525.1) using HISAT2 (v2.0.5) with default parameters. Read counts aligned to each gene were converted into FPKM (Fragments Per Kilobase of transcript sequence per Million base pairs sequenced) values. Differential gene expression analysis was conducted using DESeq2 (v1.20.0), with genes exhibiting adjusted *p*-values (*p*adj) < 0.05 and absolute log2 fold change (|log2FC|) > 2 defined as differentially expressed genes (DEGs).

### 4.3. Statistical Analysis

All statistical analyses were conducted using IBM SPSS Statistics 27.0 (IBM Corporation, Armonk, NY, USA). For physiological and biochemical indices—including seedling growth traits (seedling height, base diameter, shoot length, leaf count on shoots, root length, root diameter, biomass, survival rate), antioxidant enzyme activities (SOD, POD, CAT), soluble protein, soluble sugars, and hormone levels (IAA)—descriptive statistics (means ± standard deviations) were calculated for each graft combination and time point. Differences between groups were analyzed using one-way analysis of variance (ANOVA) to assess overall variation, followed by Duncan’s new multiple range test for pairwise comparisons. Statistical significance was defined as *p* < 0.05.

## 5. Conclusions

This study elucidates the key mechanisms underlying *Camellia oleifera* graft healing by comparing high- (Comb. 4: Xianglin 27 rootstock + Xianglin 210 scion) and low-compatibility (Comb. 7: Xianglin 210 rootstock + Xianglin 210 scion) combinations. High-compatibility grafts exhibited superior growth traits and survival rates, driven by rapid antioxidant activation (significantly elevated SOD/POD/APX activities, *p* < 0.05) and auxin synthesis (upregulated *YUCCA*/*TAA1*), which minimized oxidative damage and promoted callus formation. Transcriptomic analysis revealed 3781 differentially expressed genes, with WGCNA highlighting co-expressed genes, including stress-responsive hubs (*Hsp70*, *DnaJ*) and sugar transporters (*ABCG*), annotated to be critical for healing. The findings establish that rapid redox regulation, coordinated hormone dynamics, and efficient carbon allocation are pivotal for graft success. Comb. 4 (Xianglin 27 rootstock) is recommended for its compatibility, while the identified molecular markers (e.g., *DnaJ*, *FKBP19*) offer targets for breeding high-affinity rootstocks. This work advances the molecular understanding of woody plant grafting and provides a framework for optimizing *C. oleifera* propagation.

## Figures and Tables

**Figure 1 plants-14-02432-f001:**
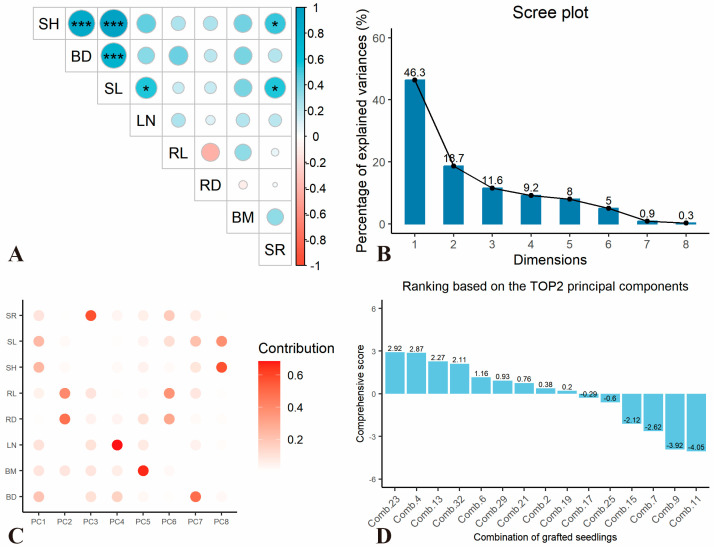
Principal component analysis of growth-related traits in different grafting combinations. (**A**) The correlation analysis of growth-related indicators used for the comprehensive evaluation of graft compatibility among different combinations; (**B**) variance contribution rate of each principal component; (**C**) correlation between seedling growth-related indicators and principal components; and (**D**) comprehensive evaluation score of each combination. The size of circle in (**A**) represented correlation coefficient, while * represented *p* < 0.05, *** represented *p* < 0.001.

**Figure 2 plants-14-02432-f002:**
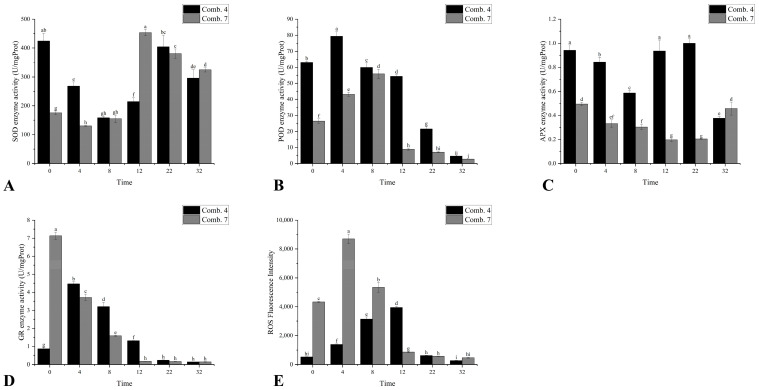
Antioxidant-related indicators of different grafting combinations in the healing process. (**A**) the enzyme activity of SOD; (**B**) the enzyme activity of POD; (**C**) the enzyme activity of APX; (**D**) the enzyme activity of GR; and (**E**) the reactive oxygen species (ROS) fluorescence intensity. According to Duncan’s multiple range test, each bar is labeled with both uppercase and lowercase letters to indicate statistical significance. Lowercase letters denote significant differences between the means of different combinations at the same time point (for example, ‘a’ differing from ‘b’ signifies (*p* < 0.05)). Vertical bars represent the standard deviation of the mean (*n* = 3).

**Figure 3 plants-14-02432-f003:**
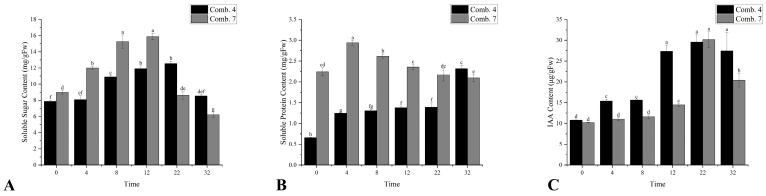
Growth-related indicators of different grafting combinations in the healing process. (**A**) the content of soluble sugar; (**B**) the content of soluble protein; and (**C**) the content of IAA. According to Duncan’s multiple range test, each bar is labeled with both uppercase and lowercase letters to indicate statistical significance. Lowercase letters denote significant differences between the means of different combinations at the same time point (for example, ‘a’ differing from ‘b’ signifies (*p* < 0.05)). Vertical bars represent the standard deviation of the mean (*n* = 3).

**Figure 4 plants-14-02432-f004:**
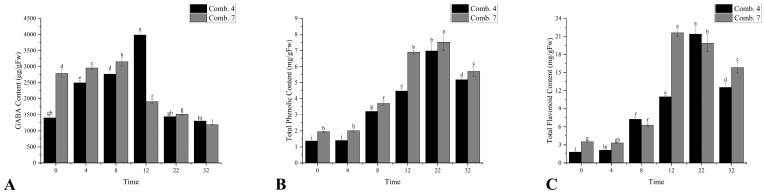
Metabolism-related indicators of different grafting combinations in the healing process. (**A**) the content of gamma-aminobutyric acid (GABA); (**B**) the content of total phenol; and (**C**) the content of total flavonoid. According to Duncan’s multiple range test, each bar is labeled with both uppercase and lowercase letters to indicate statistical significance. Lowercase letters denote significant differences between the means of different combinations at the same time point (for example, ‘a’ differing from ‘b’ signifies (*p* < 0.05)). Vertical bars represent the standard deviation of the mean (*n* = 3).

**Figure 5 plants-14-02432-f005:**
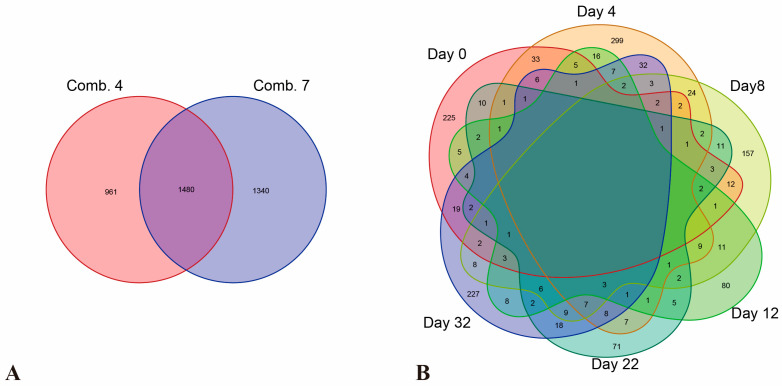
Venn diagram of DEGs identified during the healing process of different grafting combinations (**A**) and DEGs identified in the comparison of different grafting combinations at different stages during the healing process (**B**).

**Figure 6 plants-14-02432-f006:**
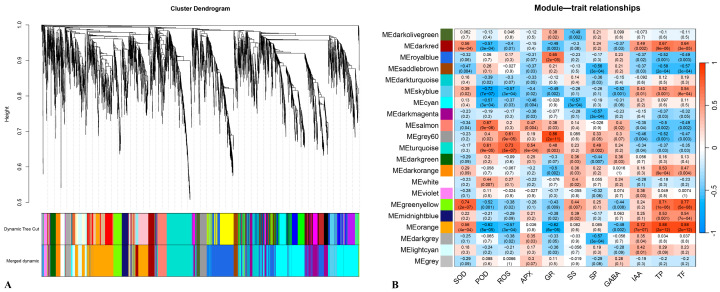
WGCNA of DEGs between different combinations during its healing process. (**A**) Hierarchical cluster tree showing co-expression modules identified by WGCNAs; each branch in the tree represents an individual gene. (**B**) Correlation between module eigengenes and physiological/biochemical indicators during the healing process post-grafting.

**Figure 7 plants-14-02432-f007:**
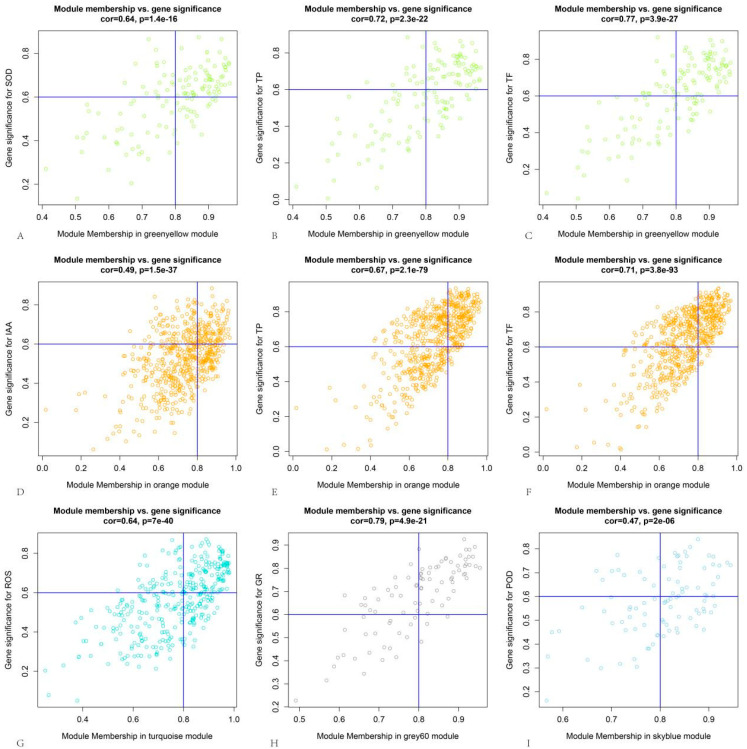
Correlation analysis between coexpression gene modules and traits: (**A**) correlation analysis between green-yellow module and SOD activity; (**B**) correlation analysis between green-yellow module and TP content; (**C**) correlation analysis between green-yellow module and TF content; (**D**) correlation analysis between orange module and IAA content; (**E**) correlation analysis between orange module and TP content; (**F**) correlation analysis between orange module and TF content; (**G**) correlation analysis between turquoise module and ROS fluorescence intensity; (**H**) correlation analysis between grey60 module and GR activity; and (**I**) correlation analysis between sky-blue module and POD acitivity. The blue line in the each figure represents the threshold of the corresponding coordinate axis.

**Figure 8 plants-14-02432-f008:**
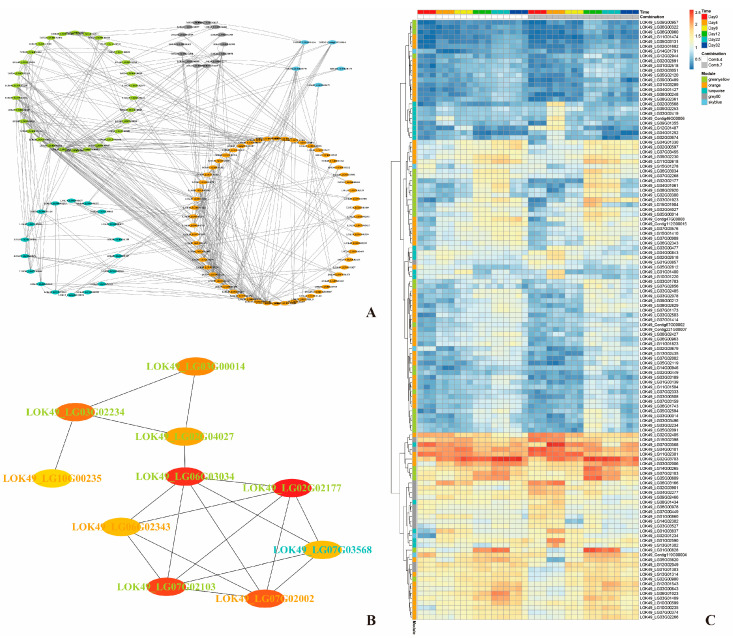
The PPI network and heatmap analysis of DEGs associated with physiological indicators: (**A**) gene co-expression networks of DEGs associated with physiological/biochemical indicators during the healing process post-grafting; (**B**) correlation network of top 10 nodes associated with physiological/biochemical indicators during the healing process post-grafting; and (**C**) heatmap showing the expression profiles of DEGs in modules significantly correlated to physiological/biochemical indicators during the healing process post-grafting. The color depth of each cell represents the gene expression level by log_2_(fpkm + 1).

**Table 1 plants-14-02432-t001:** Growth-related trait information for different grafting combinations.

No.	SH(cm)	BD (mm)	SL (cm)	LN	RL (cm)	RD (cm)	BM (g)	SR (%)
Comb. 2	15.12 ± 0.45 d	1.96 ± 0.06 abcde	5.43 ± 0.92 bcd	4.78 ± 0.19 ab	14.7 ± 1.91 ab	3.29 ± 0.33 bcd	5.16 ± 0.08 def	0.74 ± 0.02 ab
Comb. 4	19.82 ± 0.62 a	2.15 ± 0.14 ab	9.69 ± 1.24 a	4.78 ± 0.51 ab	15.27 ± 0.93 ab	3.10 ± 0.09 cde	4.98 ± 0.68 def	0.83 ± 0.04 a
Comb. 6	13.33 ± 0.49 ef	2.02 ± 0.20 abcd	4.56 ± 0.26 cde	4.67 ± 0.58 ab	15.75 ± 1.82 ab	3.13 ± 0.06 cde	6.36 ± 0.45 abc	0.45 ± 0.16 c
Comb. 7	12.19 ± 0.53 gh	1.80 ± 0.23 def	4.49 ± 0.71 cde	4.89 ± 0.38 ab	13.03 ± 0.41 bcd	3.69 ± 0.09 a	4.28 ± 0.46 f	0.74 ± 0.05 ab
Comb. 9	10.07 ± 0.62 jk	1.64 ± 0.09 f	2.26 ± 0.38 g	3.67 ± 0.34 c	12.00 ± 0.98 cd	3.04 ± 0.10 defg	4.69 ± 0.21 ef	0.71 ± 0.07 ab
Comb. 11	11.45 ± 0.39 hi	1.72 ± 0.04 ef	4.26 ± 0.53 cdef	4.45 ± 0.39 abc	11.10 ± 1.97 d	3.43 ± 0.27 abc	5.10 ± 0.62 def	0.42 ± 0.09 c
Comb. 13	13.34 ± 0.58 ef	1.94 ± 0.06 abcde	5.72 ± 0.42 bc	5.00 ± 0.33 a	15.37 ± 1.16 ab	2.93 ± 0.06 efg	6.84 ± 0.36 a	0.74 ± 0.08 ab
Comb. 15	9.85 ± 0.56 k	1.72 ± 0.09 ef	3.35 ± 0.34 efg	4.89 ± 0.19 ab	14.10 ± 1.87 abc	2.58 ± 0.25 hi	2.84 ± 0.19 g	0.42 ± 0.11 c
Comb. 17	16.58 ± 0.43 c	2.11 ± 0.32 abc	6.63 ± 2.63 b	5.00 ± 0.33 a	13.11 ± 1.46 bcd	3.58 ± 0.15 ab	5.41 ± 0.61 cde	0.53 ± 0.1 c
Comb. 19	11.71 ± 0.17 ghi	1.73 ± 0.05 ef	4.03 ± 0.64 cdef	4.89 ± 0.19 ab	14.13 ± 2.09 abc	2.72 ± 0.36 ghi	5.90 ± 0.42 abcd	0.70 ± 0.07 ab
Comb. 21	12.60 ± 1.02 fg	1.79 ± 0.01 def	5.45 ± 0.66 bcd	4.89 ± 0.38 ab	15.18 ± 1.33 ab	2.89 ± 0.13 efgh	5.78 ± 0.47 bcd	0.59 ± 0.11 bc
Comb. 23	18.44 ± 0.54 b	2.16 ± 0.07 a	9.35 ± 0.85 a	4.89 ± 0.77 ab	14.14 ± 0.70 abc	3.06 ± 0.19 defg	6.46 ± 0.58 ab	0.78 ± 0.05 a
Comb. 25	10.85 ± 0.44 ij	1.96 ± 0.08 abcde	2.68 ± 0.20 fg	4.00 ± 0.33 bc	15.49 ± 0.92 ab	2.75 ± 0.04 fghi	4.62 ± 0.31 ef	0.44 ± 0.15 c
Comb. 29	13.45 ± 0.32 ef	1.87 ± 0.08 cdef	3.90 ± 0.29 def	4.55 ± 0.39 ab	16.40 ± 1.27 a	3.09 ± 0.17 cdef	5.82 ± 0.18 bcd	0.58 ± 0.11 bc
Comb. 32	14.15 ± 0.87 e	1.88 ± 0.21 bcdef	5.57 ± 0.31 bcd	4.67 ± 1.00 ab	14.5 ± 0.59 abc	2.45 ± 0.04 i	6.57 ± 1.21 ab	0.73 ± 0.09 ab

Lowercase letters denote significant differences between the means of different combinations at the same time point (for example, ‘a’ differing from ‘b’ signifies (*p* < 0.05)).

**Table 2 plants-14-02432-t002:** Statistics of differentially expressed genes varied in different stage during the healing process.

Combinations of Grafted Seedlings	Regulation	Day 4 vs. Day 0	Day 8 vs. Day 0	Day 12 vs. Day 0	Day 22 vs. Day 0	Day 32 vs. Day 0
Comb. 4	up	526	448	407	453	490
	Down	644	650	543	620	684
	Total	1170	1098	950	1073	1174
Comb. 7	up	557	530	532	377	343
	Down	847	431	626	566	501
	Total	1404	961	1158	943	844

**Table 3 plants-14-02432-t003:** Statistics of differentially expressed genes varied in two conbinitions during the healing process.

Comb. 4 vs. Comb. 7	Upregulated	Downregulated	Total
Day 0	130	216	346
Day 4	291	185	476
Day 8	117	169	286
Day 12	78	101	179
Day 22	115	65	180
Day 32	222	162	384

**Table 4 plants-14-02432-t004:** DEGs identified in the tryptophan metabolism pathway at 4 DPG.

ID	log2FC	Pathway	Knum	Gene_Symbol
LOK49_LG04G01187	3.08	Tryptophan metabolism	K00128	ALDH
LOK49_LG03G03745	2.20	Tryptophan metabolism	K01426	amiE
LOK49_LG02G03861	5.60	Tryptophan metabolism	K01426	amiE
LOK49_LG13G02424	2.99	Tryptophan metabolism	K11816	YUCCA
LOK49_LG14G00862	−2.15	Tryptophan metabolism	K11820	UGT74B1
LOK49_LG06G02130	2.45	Tryptophan metabolism	K11820	UGT74B1
LOK49_LG10G00377	7.06	Tryptophan metabolism	K11820	UGT74B1
LOK49_LG05G01623	7.44	Tryptophan metabolism	K11820	UGT74B1
LOK49_LG01G01027	−2.38	Tryptophan metabolism	K11821	ST5A
LOK49_LG14G01022	2.15	Tryptophan metabolism	K16903	TAA1

**Table 5 plants-14-02432-t005:** DEGs associated with sugar translocation at 4 DPG in different combinations.

ID	Description	Comb. 4 log2FC	Comb. 7 log2FC
LOK49_LG01G02312	ABC transporter D family member	−2.62	−3.72
LOK49_LG01G02277	ABC transporter G family member 11-like	2.92	2.95
LOK49_LG10G01469	ABCG family. PDR (TC 3.A.1.205) subfamily	−3.00	−3.68
LOK49_LG01G02478	ABC transporter G family member	2.60	2.69
LOK49_LG01G02285	ABC transporter G family member 11-like	3.01	2.53
LOK49_LG04G02801	ABC transporter C family member	2.07	2.07
LOK49_LG01G03892	ABC transporter B family member	−4.19	−4.22
LOK49_LG03G02380	UDP–galactose UDP–glucose transporter	−2.82	−2.34
LOK49_LG12G01226	Sugar transporter (TC 2.A.1.1) family	2.28	2.04
LOK49_LG01G03893	ABC transporter B family member	−4.39	−4.18
LOK49_LG03G02406	UDP–galactose UDP–glucose transporter	−2.28	−2.30
LOK49_LG03G01565	UDP–sugar transporter	−4.64	−5.82
LOK49_LG13G01063	ABC transporter C family member 10-like	2.08	−4.93
LOK49_LG01G02312	ABC transporter D family member	−2.62	−3.72

**Table 6 plants-14-02432-t006:** Hub genes from WGCNA analysis of DEGs during graft healing.

Gene_ID	Module	Description
LOK49_LG02G02177	green-yellow	Heat shock 70 kDa protein BIP1
LOK49_LG07G02002	orange	Chaperone protein dnaJ C76
LOK49_LG03G00014	green-yellow	Peptidyl-prolyl cis-trans isomerase FKBP19
LOK49_LG03G02234	green-yellow	Syntaxin-71
LOK49_LG07G03568	turquoise	Chaperone protein dnaJ 11
LOK49_LG06G02343	orange	Chaperone protein dnaJ 20
LOK49_LG06G03034	green-yellow	Heat shock 70 kDa protein 8
LOK49_LG07G02103	green-yellow	Heat shock protein Hsp40-2
LOK49_LG02G04027	green-yellow	Uncharacterized sugar kinase slr0537
LOK49_LG10G00235	orange	Very-long-chain 3-oxoacyl-CoA reductase 1

**Table 7 plants-14-02432-t007:** Design of grafting combinations.

No.	Scion	Rootstock	Characteristics of Rootstock
Comb. 2	Xianglin 210	Xianglin 1	National key promoted improved varieties, high yield, large fruit size, stress resistance
Comb. 4	Xianglin 210	Xianglin 27	National key promoted improved varieties, high yield, thin pericarp, stress resistance
Comb. 6	Xianglin 210	Xianglin 97	National key promoted improved varieties, high yield, thin pericarp, stress resistance
Comb. 7	Xianglin 210	Xianglin 210	National key promoted improved varieties, high yield, large fruit size, stress resistance
Comb. 9	Xianglin 210	Xianglin 82	Excellent hybrid progeny germplasm
Comb. 11	Xianglin 210	Guoyou 12	High-yield new varieties
Comb. 13	Xianglin 210	Guoyou 13	High-yield new varieties
Comb. 15	Xianglin 210	Guoyou 14	High-yield new varieties
Comb. 17	Xianglin 210	Guoyou 15	High-yield new varieties
Comb. 19	Xianglin 210	Zhongzhi 3	Excellent hybrid progeny germplasm
Comb. 21	Xianglin 210	Dezi 1	Provincial key promoted improved varieties, high yield, large fruit size, stress resistance
Comb. 23	Xianglin 210	Xianglin183	Excellent hybrid progeny germplasm
Comb. 25	Xianglin 210	Xianglin331	Excellent hybrid progeny germplasm
Comb. 29	Xianglin 210	Zhongzhi 2	Excellent hybrid progeny germplasm, high yield, large fruit size, stress resistance
Comb. 32	Xianglin 210	Youxian	Excellent local varieties

## Data Availability

The raw RNA-seq data generated in this study have been deposited in the NCBI Sequence Read Archive (SRA) under accession number PRJNA1272790 (https://www.ncbi.nlm.nih.gov/bioproject/PRJNA1272790 accessed on 6 June 2025).

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
