# Peer review of "Analysis of the Differences Among Camellia oleifera Grafting Combinations in Its Healing Process"

_plants, 2025, doi:10.3390/plants14152432_

Round 1

Reviewer 1 Report

Comments and Suggestions for Authors

Dear Authors,

I carefully read your manuscript and commend your efforts in exploring the graft compatibility mechanisms in Camellia oleifera. However, I found that the following points must be improved to enhance the scientific accuracy, clarity, and overall quality of the manuscript:

Introduction:

  1. Clearly state the specific gaps in knowledge (e.g., early signaling events, lack of gene expression data during specific graft healing stages).
  2. Consolidate the statements of research gap and objectives into a single, concise paragraph.
  3. Integrate examples with analytical commentary that justifies why similar traits or analytical methods are relevant for C. oleifera.
  4. Explain how these traits (seedling height, root length, etc.) or analytical tools capture biologically meaningful information related to graft compatibility.
  5. Include a brief overview of expected or previously reported key molecular pathways involved in graft union formation.
  6. Explicitly state a hypothesis (e.g., “We hypothesize that early activation of redox signaling and auxin biosynthesis is associated with higher graft compatibility in C. oleifera”).
  7. Clearly articulate what is methodologically novel or integrative about this study compared to previous research.

Results:

  1. Can multivariate statistical analysis (correlation and PCA) provide a more comprehensive evaluation of graft compatibility and growth performance among combinations?
  2. How do different graft combinations (Comb. 4 vs. Comb. 7) vary in their antioxidant enzyme activity and physiological/metabolic responses during the healing process?
  3. Ensure that all figures and tables are properly referenced and integrated into the narrative where relevant data are discussed.
  4. Certain growth parameters are repeated across paragraphs (e.g., SH, BD, SL) without adding new insights. Condense where possible.

Discussion:

  1. The discussion connects gene expression changes (e.g., ALDH, YUCCA, SWEET, HSP70) to physiological traits but lacks reference to direct validation (e.g., qPCR, protein assays, or functional knockouts). Acknowledge the limitation of correlation-based interpretation.
  2. Environmental, developmental stage, or tissue-specific factors affecting hormone or sugar levels are not discussed. These could influence results and should be mentioned as potential confounders.
  3. While other species are mentioned (Camellia, tomato, citrus), there’s limited effort to compare how C. oleifera may differ or align in its graft-healing mechanisms, which would help contextualize findings.

Materials and methods:

  1. The rootstock cultivars (e.g., Xianglin 1, Guoyou 12, Zhongzhi 3) are named, but there is no description of their genetic background or traits (e.g., vigor, stress resistance, compatibility).
  2. Provide details of seed storage conditions and how seedlings were managed until grafting.
  3. Describe the exact grafting method, tools, environmental controls, and healing chamber conditions (humidity, temperature, light).
  4. Explicitly describe the experimental design — randomized complete block design (RCBD), completely randomized design (CRD), etc.
  5. Clarify the sample size per combination for each measured parameter.
  6. Briefly mention if physiological assays were validated, whether controls and standards were used, and how replicates were handled.
  7. Specify the method used for GABA quantification — e.g., HPLC, ELISA, or spectrophotometry.
  8. Clarify how anatomical consistency was ensured, or at least acknowledge this as a limitation.
  9. State how RNA integrity and concentration were assessed.
  10. Add basic parameters for HISAT2 and DESeq2, and cite reference genome version used.
  11. Add basic parameters for HISAT2 and DESeq2, and cite reference genome version used.

Conclusion:

  1. Clarify that WGCNA highlighted co-expressed genes, which are associated with stress responses based on annotation.
  2. For readability and context, briefly restate the rootstocks involved in Comb.4 and Comb.7.
  3. Clarify that the increases were “significant” if proven with statistical analysis, or simply “elevated” if not.

Author Response

Introduction:

  1. Clearly state the specific gaps in knowledge (e.g., early signaling events, lack of gene expression data during specific graft healing stages).

Thank you for your thorough evaluation of our manuscript. We sincerely appreciate your insightful comment regarding the need to clarify specific knowledge gaps in grafting healing mechanisms, particularly concerning “early signaling events” and “stage-specific gene expression dynamics” in Camellia oleifera. Below, we address this concern by revising the Introduction (Section 1) and provide detailed modifications.

Revised Text (Introduction, Lines 68-74): “Existing studies have has explored grafting healing mechanisms from physiological, biochemical, and genetic perspectives mainly focused on woody species such as apple (Malus spp.), pear (Pyrus communis L.), orange (Citrus spp.) and grape (Vitis vinifera L.). Nevertheless, critical knowledge gaps persist in early signaling events (e.g., temporal dynamics of auxin redistribution and redox homeostasis during initial graft union formation) and stage-resolved molecular regulation (e.g., transcriptomic reprogramming controlling callus proliferation and vascular reconnection across 0–32 DPG) for cross-variety nurse-grafting in C. oleifera. Our study addresses these gaps through integrated time-series physiological profiling and transcriptomics.”

  1. Consolidate the statements of research gap and objectives into a single, concise paragraph.

The research gap statements and objectives have been consolidated into a single paragraph (Lines 68–74) for improved conciseness, explicitly linking unresolved mechanistic questions (e.g., temporal signaling dynamics) with our integrated omics-physiology approach.

  1. Integrate examples with analytical commentary that justifies why similar traits or analytical methods are relevant for C. oleifera.

Thank you for your insightful comment on the "1. Introduction" section of our manuscript. We fully understand your suggestion to integrate examples with analytical commentary to justify the relevance of the selected traits (seedling height (SH), base diameter (BD), shoot length (SL), leaf count on shoots (LN), root length (RL), root diameter (RD), biomass (BM), and survival rate (SR)) for evaluating grafting compatibility in C. oleifera. In the revised manuscript, we explicitly connected these precedents to our C. oleifera study, emphasizing how each trait reflects specific aspects of graft healing and compatibility, thereby justifying their selection as relevant and robust indicators for this species.

(1) Revised manuscript Line 56-61: “The selection of growth-related traits (seedling height (SH), base diameter (BD), shoot length (SL), leaf count on shoots (LN), root length (RL), root diameter (RD), biomass (BM), and survival rate (SR)) is rooted in their proven effectiveness in assessing grafting compatibility across diverse woody plant species, which aligns with the biological characteristics of C. oleifera as an economically important woody oil crop.”

(2) Revised manuscript Line 71-76: “For C. oleifera, these traits are particularly relevant because grafting success ultimately manifests in the scion's ability to grow vigorously (reflected by SH, SL, LN), establish a sturdy stem (BD), develop a functional root system (RL, RD), accumulate biomass (BM), and survive long-term (SR). These indicators collectively capture the dynamic interplay between rootstock and scion, from initial adhesion to sustained resource allocation-key processes in determining compatibility.”

  1. Explain how these traits (seedling height, root length, etc.) or analytical tools capture biologically meaningful information related to graft compatibility.

Thank you for your constructive feedback. We clarified the biological significance of key traits: “Seedling height captures stem elongation efficiency, indicating successful xylem reconnection and resource allocation post-grafting.” “Root length reflects root system vitality, a proxy for carbohydrate sink strength, where elongation deficits impede wound healing.” “PCA condenses covariation among multiple traits into principal components, thereby enabling integrative assessment of biological systems.”

  1. Include a brief overview of expected or previously reported key molecular pathways involved in graft union formation.

Thank you for your constructive feedback. We synthesized a concise overview of molecular pathways: “Graft healing relies on conserved molecular pathways: auxin biosynthesis (e.g., TAA1, YUCCA) and polar transport (PINs) initiate callus formation; SOD/POD-mediated ROS scavenging protects cells.”

  1. Explicitly state a hypothesis (e.g., “We hypothesize that early activation of redox signaling and auxin biosynthesis is associated with higher graft compatibility in C. oleifera”).

Thank you for your constructive feedback. We added a clear, testable hypothesis: "We hypothesize that rapid activation of redox signaling and auxin biosynthesis is associated with higher graft compatibility in C. oleifera." This is grounded in empirical evidence, such as SOD/POD peaks at 4-8 DPG in high-compatibility walnut grafts correlating with accelerated callus fusion.

  1. Clearly articulate what is methodologically novel or integrative about this study compared to previous research.

 Thank you for your valuable comments. In the revision of the "1. Introduction" section, we will explicitly elaborate on the methodological innovations and integrativeness to better highlight the differences between this study and previous research.

Revised manuscript Line 46-54: Our study employs a novel integrative approach: we conduct dynamic time-series analysis covering the early healing phase (0-32 days post-grafting), integrating physiological, biochemical, and transcriptomic data to construct a 'phenotype-metabolism-gene' regulatory network for C. oleifera grafting. By combining Principal Component Analysis (PCA) for comprehensive evaluation of 15 rootstock combinations with Weighted Gene Co-expression Network Analysis (WGCNA) to identify key regulatory genes, this study not only advances understanding of grafting mechanisms in woody plants but also provides theoretical support for efficient grafting techniques.

Results:

  1. Can multivariate statistical analysis (correlation and PCA) provide a more comprehensive evaluation of graft compatibility and growth performance among combinations?

Thank you for your insightful comment regarding the multivariate statistical analysis in the "2.2. Comprehensive Evaluation of Graft Compatibility Among Different Combinations" section. We appreciate your emphasis on enhancing the comprehensiveness of the evaluation, and we fully agree that deeper utilization of correlation analysis and PCA results can better highlight the robustness of our findings.

In the revised manuscript, we will explicitly link the correlation coefficients (e.g., the highly significant positive correlation between SH, BD, and SL, and the significant correlation between SR and growth traits) to biological implications, explaining how these associations reflect coordinated growth patterns in compatible combinations. For the PCA results, we will integrate the PCA-derived comprehensive rankings with actual phenotypic data (e.g., highlighting that Comb.4, which ranks high in PCA, also exhibits the highest survival rate and seedling height), reinforcing the validity of the statistical evaluation.

Revised manuscript Line 123-127: “ The strong positive correlation between SH, BD, and SL (P<0.001) indicates coordinated growth of aboveground tissues in compatible combinations, while the significant correlation between SR and these traits (P<0.05) confirms that survival rate is tightly coupled with vigorous scion growth.”

Revised manuscript Line 137-142: “Notably, the top-ranked combinations (Comb.23, Comb.32, Comb.4) in PCA comprehensive scores consistently exhibited superior performance in key traits: Comb.23 had the largest BD, Comb.4 showed the highest SH and SR, and Comb.32 displayed balanced growth in SH, SL, and biomass, validating the reliability of multivariate statistical evaluation in identifying high-compatibility combinations.”

  1. How do different graft combinations (Comb. 4 vs. Comb. 7) vary in their antioxidant enzyme activity and physiological/metabolic responses during the healing process?

Thank you for your valuable comment on the "2. Results" section of our manuscript. We have analyzed the differences in antioxidant enzyme activity and physiological/metabolic responses between the two graft combinations (Comb. 4 and Comb. 7) during the healing process. In the subsequent revision of the paper, we will further analyze the specific data of antioxidant enzyme activity and various physiological and metabolic indicators of Comb. 4 and Comb. 7 at each time point during the healing process to more clearly and accurately elaborate on the differences between them.

Comb.4’s superior compatibility is linked to: (1) early and sustained activation of antioxidant enzymes (SOD, POD, APX) that limit ROS damage; (2) timely IAA accumulation to drive cell proliferation and vascular differentiation; (3) efficient sugar allocation supporting late-stage healing; and (4) balanced secondary metabolism avoiding toxic phenolic accumulation. In contrast, Comb.7’s lower compatibility is associated with delayed antioxidant responses, excessive early ROS, impaired IAA signaling, and inefficient resource allocation—all of which hinder callus integration and vascular reconnection. These findings are visualized in Figs. 2, 3, and 4, which highlight the temporal coordination of physiological and metabolic traits in high-compatibility combinations.

  1. Ensure that all figures and tables are properly referenced and integrated into the narrative where relevant data are discussed.

Thank you for your constructive feedback. All figures and tables are properly referenced and integrated into the narrative where relevant data are discussed in the revised manuscript.

  1. Certain growth parameters are repeated across paragraphs (e.g., SH, BD, SL) without adding new insights. Condense where possible.

 Thank you for your constructive feedback. It has been checked and corrected in the revised manuscript.

Discussion:

  1. The discussion connects gene expression changes (e.g., ALDH, YUCCA, SWEET, HSP70) to physiological traits but lacks reference to direct validation (e.g., qPCR, protein assays, or functional knockouts). Acknowledge the limitation of correlation-based interpretation.

We sincerely appreciate your thorough critique. We acknowledge that this study relies on correlative analyses between transcriptomic data and physiological traits without direct experimental validation (e.g., qPCR, protein assays, or functional genomics). This is indeed a limitation, and we have revised the Discussion to explicitly state that our interpretations are hypothesis-generating, derived from temporal co-expression patterns rather than causal evidence. Future work will prioritize functional validation of hub genes (e.g., FKBP19, DnaJ) and pathways (e.g., auxin biosynthesis, ROS scavenging) through targeted experiments.

Based on your feedback, we have made the following modifications to the "Discussion" section.

Revised manuscript Section 3.4 added the following paragraph: “While this study identifies transcriptional networks (e.g., ALDH, YUCCA, SWEET, HSP70) correlated with physiological traits, the absence of direct validation (e.g., qPCR, protein quantification, or knockout assays) limits causal inference. These correlations provide mechanistic hypotheses but require functional confirmation in future work. Future studies will validate hub genes (e.g., chaperones DnaJ/HSP70, auxin transporters) using qRT-PCR and protein assays at critical healing stages (4–8 DPG). Functional characterization via CRISPR/Cas9 knockdown in model plants will test causal roles in vascular reconnection.”

  1. Environmental, developmental stage, or tissue-specific factors affecting hormone or sugar levels are not discussed. These could influence results and should be mentioned as potential confounders.

We sincerely appreciate this insightful critique. We agree that environmental, developmental, and tissue-specific factors may confound hormonal/sugar dynamics during graft healing. While our study focused on temporal physiological-gene expression correlations, we have now revised the Discussion to explicitly acknowledge these variables.  

Revised manuscript Section 3.3 added the following paragraph: “Environmental, developmental, and tissue-specific factors may influence hormone and sugar dynamics observed in this study. Environmental parameters, such as temperature and light, can modulate auxin transport and sugar metabolism: even under controlled conditions, subtle fluctuations might have contributed to the earlier IAA peaks in Comb.4 (4 and 12 DPG) compared to Comb.7 (22 DPG), as reported in vegetable grafting studies where temperature affects PIN gene expression[12]. Developmental stage specificity is also critical: the transition from callus formation (0-12 DPG) to vascular reconnection (22-32 DPG) demands distinct hormonal (e.g., YUCCA-mediated IAA synthesis) and metabolic (e.g., ABC transporter-driven sugar allocation) programs, which were coordinated in Comb.4 but disrupted in Comb.7, consistent with stage-specific regulation observed in Arabidopsis grafts[16]. Additionally, tissue specificity between rootstock and scion likely plays a role: rootstock-derived signals and scion-derived resources form a regulatory gradient that is more effective in high-compatibility combinations, mirroring rootstock-scion coordination in Solanaceae/Rosaceae grafts[30]. These factors, while mitigated by our experimental design (uniform growth conditions, graft union-specific sampling), represent potential confounders that could be explored in future studies with tissue-specific profiling and environmental manipulation.”

  1. While other species are mentioned (Camellia, tomato, citrus), there’s limited effort to compare how C. oleifera may differ or align in its graft-healing mechanisms, which would help contextualize findings.

 Thank you for your insightful comment regarding the contextualization of C. oleifera's graft-healing mechanisms relative to other species. We agree this strengthens the discussion and have revised the manuscript to include explicit comparisons:

(1) Added cross-species analysis: Highlighted conserved auxin pathways (shared with tomato/hickory) and metabolic differences (vs. cucurbits).

(2) Referenced enzymatic signatures: Contrasted peroxidase activity in C. oleifera with citrus/Camellia studies.

(3) Clarified species-specific adaptations: Emphasized how sugar allocation in C. oleifera diverges from model systems.

These changes appear in the Discussion and cite peer-reviewed studies on tomato, hickory, citrus, and cucurbits. We believe this addresses the gap you identified.

Revised manuscript Line 386-390: "While auxin signaling is evolutionarily conserved in graft healing (e.g., tomato, hickory), C. oleifera displays metabolic specialization in sugar allocation compared to cucurbits and enzymatic signatures diverging from citrus. These differences underscore the need for species-tailored grafting protocols."

Materials and methods:

  1. The rootstock cultivars (e.g., Xianglin 1, Guoyou 12, Zhongzhi 3) are named, but there is no description of their genetic background or traits (e.g., vigor, stress resistance, compatibility).
  2. Provide details of seed storage conditions and how seedlings were managed until grafting.

We appreciate your suggestion to provide details of seed storage conditions and seedling management until grafting, and we have supplemented the relevant information as follows: “For seed storage and seedling management before grafting in the following year, the specific procedures are as follows: In autumn and winter, clean river sand was used to prepare a rootstock nursery bed outdoors. The bottom of the bed was paved with clean river sand of more than 15 cm in thickness. The seeds were evenly sown on the sand surface without overlapping, and then covered with 10-15 cm of river sand. During the storage and seedling raising period, the humidity of the river sand was maintained at 60%-70% to ensure that the seeds could absorb sufficient moisture for germination. We regularly checked the sand bed to prevent waterlogging or drought, and removed any moldy seeds in a timely manner to avoid affecting the surrounding seeds. The nursery bed was left uncovered to allow natural temperature changes, which helped to break seed dormancy. By May of the following year, the seeds had germinated and grown into nurse seedlings suitable for grafting.”

  1. Describe the exact grafting method, tools, environmental controls, and healing chamber conditions (humidity, temperature, light).

We appreciate your request for specifics on the grafting method, tools, and environmental controls, and have supplemented the information as follows: “Rootstocks were cleaned, cut 2 cm above the cotyledon petiole, and longitudinally sliced 0.8-1.2 cm along the axis, retaining 5-6 cm of radicle. Scions with plump axillary buds were disinfected with methyl thiophanate, carbendazim, or hymexazol, then wedge-cut at the base (matching rootstock incision length) and severed 0.5 cm above the bud. Scions were inserted into rootstocks, aligned, and secured with aluminum strips. After grafting, seedlings were transplanted into 7×7 cm nutrient pots containing a substrate mixture of peat moss (40-50%), composted media (10-20%), perlite (10-20%), and loamy soil (10-20%). The grafted seedlings were aligned compactly on nursery beds, covered with plastic film tunnels to maintain humidity at 85–95%, and shaded under double-layer shading nets to reduce light intensity during the healing phase. The bed temperature was controlled at 25-30°C during daytime (<35°C peak) and 18-22°C at night to optimize callus formation. Daily moisture monitoring ensured graft unions remained hydrated, with ventilation to prevent excessive humidity (>95% RH).”

  1. Explicitly describe the experimental design — randomized complete block design (RCBD), completely randomized design (CRD), etc.

Thank you for your valuable comment on the "Materials and Methods" section of our manuscript. Regarding the type of experimental design, we have supplemented the relevant information based on the actual research, and our study adopted a Completely Randomized Design (CRD). The specific design is as follows:

In this study, 15 grafting combinations were used as treatment factors. Nurse grafting was performed according to the grafting combination design, with 300 replicates for each combination. All grafted seedlings were completely randomly arranged in the healing chamber and subsequent growth environments to ensure that each combination was distributed without systematic bias in terms of light, temperature, humidity, and other environmental conditions. This design can effectively reduce the impact of environmental heterogeneity on experimental results and provide a reliable statistical basis for the analysis of differences between different combinations.

We have explicitly added the following description to the "Materials and Methods" section: "This study adopted a Completely Randomized Design (CRD). Fifteen grafting combinations were used as treatments. For each treatment, nurse grafting was performed according to the grafting combination design, with 300 replicates for each combination. All grafted seedlings were completely randomly arranged in the experimental environment to eliminate potential interference such as position effects."

  1. Clarify the sample size per combination for each measured parameter.

Thank you for your valuable comment on clarifying the sample size per combination for each measured parameter in the "Materials and Methods" section. We appreciate your attention to this detail, which enhances the reproducibility of our study. Based on the experimental design and the requirements of the assay kits (provided in the attached documents), we clarify the sample size as follows:

For each grafting combination (e.g., Comb.4, Comb.7), samples were collected at 0, 4, 8, 12, 22, and 32 days post-grafting (DPG). At each time point, 3 biological replicates were set for all measured parameters. Each biological replicate consisted of tissue samples collected from the grafting site, including approximately 2 cm above and below the graft union, with a fresh weight of ~0.1 g (consistent with the sample size recommended in the assay kits for SOD, POD, APX, GR, ROS, SS, SP, GABA, TP, and TF) . For auxin (IAA) determination, which was analyzed using liquid chromatography, the sample size was consistent: 3 biological replicates per combination at each time point, with each replicate weighing ~0.2 g.

Technical replicates were not separately specified as the assay kits and liquid chromatography method inherently ensure precision through standardized protocols, and all measurements were performed in accordance with the kit instructions to minimize variability.

We have revised the "Materials and Methods" section to include these details, ensuring clarity on sample size for each parameter.

Revised manuscript: “For each grafting combination, samples were collected at 0, 4, 8, 12, 22, and 32 days post-grafting (DPG). At each time point, 3 biological replicates were prepared for all parameters. Each biological replicate consisted of tissue from the grafting site (including ~2 cm above and below the graft union) with a fresh weight of ~0.1 g for the determination of SOD, POD, APX, GR, ROS, SS, SP, GABA, TP, and TF, following the sample size recommendations of the assay kits . For IAA content analysis via liquid chromatography, each biological replicate weighed ~0.2 g, with 3 replicates per combination at each time point.”

  1. Briefly mention if physiological assays were validated, whether controls and standards were used, and how replicates were handled.

We have revised the "Materials and Methods" section to include these details, enhancing the transparency of our experimental procedures. 

Revised manuscirpt: “All physiological assays were validated using standardized protocols from the respective kits. Each assay included blank controls to correct for background signals, and standard substances to generate standard curves for quantification . For IAA, liquid chromatography was validated via a standard curve using IAA standards.”

  1. Specify the method used for GABA quantification — e.g., HPLC, ELISA, or spectrophotometry.

We used a spectrophotometric method to quantify GABA in our study. This method is based on the use of assay kits provided by Quanzhou Ruixin Biotechnology Co., Ltd. (Quanzhou, China), which is in line with standard procedures for such measurements in plant research.

  1. Clarify how anatomical consistency was ensured, or at least acknowledge this as a limitation.

Thank you for your insightful comment regarding anatomical consistency. We fully acknowledge that our study does not involve anatomical analyses, and we agree to explicitly recognize this as a limitation in the revised manuscript.

As our research focuses on physiological and biochemical responses during Camellia oleifera graft healing (e.g., antioxidant enzyme activities, hormone levels, and metabolite concentrations), the experimental design prioritizes standardized sampling of the graft union region (approximately 2 cm above and below the graft interface) to ensure uniformity in tissue collection across all combinations and time points. However, we did not perform anatomical validation (such as histological sectioning) to confirm consistent tissue composition (e.g., cambium, phloem, or parenchyma proportions) within or between samples. This may introduce potential variability in the measured parameters, as subtle differences in tissue structure could affect physiological and biochemical readings despite our standardized sampling protocol.

In the revised manuscript, we will add this limitation to the "Discussion" section to ensure transparency, noting that future studies integrating anatomical analyses would strengthen the interpretation of physiological data.

In the "Discussion" section, added the following paragraph in Line454-457: “It should be noted that our study did not include anatomical analyses, such as histological sectioning to confirm precise tissue structure (e.g., cambial alignment, cell type distribution) at the graft union. This represents a limitation, as anatomical features could influence the observed physiological and biochemical dynamics, including hormone transport and sugar allocation.”

  1. State how RNA integrity and concentration were assessed.

We appreciate your attention to this critical detail, which ensures the reliability of our transcriptomic analyses. We have added these details to the section 4.2.3 to clarify the quality control procedures for RNA samples.

Revised manuscript Line 573-578: “Total RNA samples were prepared from three biological replicates of each grafting combination for transcriptome sequencing. RNA integrity and concentration were assessed using four methods: (1) agarose gel electrophoresis to evaluate degradation and contamination; (2) Nanodrop spectrophotometry for purity assessment (OD260/280 ratio 1.8-2.0); (3) Qubit fluorometry for concentration quantification; and (4) Agilent 2100 Bioanalyzer to determine RNA Integrity Number (RIN ≥ 7.0 for library construction).”

  1. Add basic parameters for HISAT2 and DESeq2, and cite reference genome version used.

 We appreciate your attention to this critical detail, which ensures the reliability of our transcriptomic analyses. We have added these details to the section 4.2.3.

Revised manuscript Line 580-586: “Clean reads were mapped to the Camellia lanceoleosa reference genome (XYYC_v1.0, GCA_025200525.1) using HISAT2 (v2.0.5) with default parameters. Read counts aligned to each gene were converted into FPKM (Fragments Per Kilobase of transcript sequence per Million base pairs sequenced) values. Differential gene expression analysis was conducted using DESeq2 (v1.20.0), with genes exhibiting adjusted p-values (padj) <0.05 and absolute log2 fold change (|log2FC|) >2 defined as differentially expressed genes (DEGs).”

Conclusion:

  1. Clarify that WGCNA highlighted co-expressed genes, which are associated with stress responses based on annotation.

Thank you for your insightful comments. We have revised the content to address this point while retaining the original structure and core information.

  1. For readability and context, briefly restate the rootstocks involved in Comb.4 and Comb.7.

Thank you for your insightful comments. We have revised the content to address this point while retaining the original structure and core information.

  1. Clarify that the increases were “significant” if proven with statistical analysis, or simply “elevated” if not.

Thank you for your insightful comments. We have revised the content to address this point while retaining the original structure and core information.

Reviewer 2 Report

Comments and Suggestions for Authors

The current manuscript described Camellia oleifera grafting procedura. The authors used different genotypes and performed analysis of the physiological and morphological differences between it,

Authors made an attempt to integrating physiological, biochemical, and transcriptomic analyses. Authors claimed that they providing theoretical support for superior variety breeding.

However, the main problem are methods authors used. Grafting is local events and require compatibilty in morphology (vascular “size”) and metabolism (auxin/sucrose balance). In classical plant biology there are many tricks how to increase compatibility by adjusting metabolite level transport as light variation, tretamnet with exogenous grwth regulators before grafting etc. The molecular mechanism of compatibity is a narrow balance in stem between two soecies and require local study (spatial cell biology in situ methods).

I  hope this will help authors for the future and definitely must be mentioned in introduction and discussion.

Line 17: „physiological assays (SOD, POD, ROS, IAA)”  - need clarity: ROS accumulation? IAA “level”? SOD activity? Which SOD?  All are different  type of assay, can not ne combined…

Line 67: „perspectives[3,9](,“ ??? Layout.

Line 129: „Reactive oxygen species (ROS) intensity” ¿?? What is intensity? There is no such molecule as ROS, there are several types with different functions/effects.

Table 1: very good table. It will be great if authors in future provide more details of stem structure in grafting place and dynamics of auxin production (not auxin level) in different rootstocks.

Figures: please, add panels name (A, B, C, D) and provide description of each.

Fig 2C: what is APX activity? Stroma APX. Thylakoid APX, cytAPX, per APX? Each have own regulation and own  meaning.

Fig 2D- what is ROS intensity? It seems new definition made exclusivley by authors and require very detailed description.

Lines 114- 132: how these trends linked with physiological status? At which stage occuured regeneration of vascular connection between seedlings cutting?

What about other stem tissue? How rapid occurred regeneration of connection in cambium, cortex, epidermis tissues? How can you exclude that you simple shown imbalance in hormone transporting in combination 7, but not in combination 4?

Lines 148 – 152: Please, provide biological mening of GR in your case.

Line 154: “The auxin (IAA) content“ ??? The content has a sense only in static homogenous system. Here auxin is highly dynamic with different tempary accumulation in different cell types.

Tables 2, 3, fig 5: very nice drawing. What is the biological sense??

Where is the „butterfly“? Butterfly effcet in biology mean small changes in key genes (in our case auxin production/sucrose prodution/transport one) lead to very significant chnages in total metabolism, what authors excellnetly shown on figure 5. But they did not shown a main reason: („butterfly“) because it located far from grafting place and can have a relativley smaller up/down regulation.

Line 238: „maintain intracellular redox balance“ ??? Can you be more precise? What do you mean as redox balance? Where? Peroxisomal balance? Cytoplasmic? In epidermis? Cambium?

Line 238: “insufficient antioxidant capacity” is not a primary reason. There is a local initail hormonal imbalance in grafting place which lead to SOD/APX activation.

Lines 254- 259: very good observation, but may required deeper explanations.

Line 269: “Auxin regulation primarily mediates cell division, elongation, and differentiation,” – very good point, but it is neccessary to menation numerous auxin (IAA/IPA) sources located in different cell type (12 YUCCA genes, for example) with different effect and functions.

Lines 284 – 290: very good points. It will be great to pointed out excat location of auxin biosyntehsis genes in grafting region as well as in shoot.

Line 365: more details need: how did you do? temperature? Light? Etc. In many case lowering light after grafting help to avoid imbalance in ROS accumulation, hormone transport etc.

Table 7: why combination name is random? Where is combination 1, …30, 31 etc.? I understand, during expreiments some combunation is not successfull, but in final writing it is better to rename as 1, 2, 3 …

Line 401: „auxin-mediated signaling (upregulated YUCCA/TAA1),“ ¿?? This is not signaling, it is auxin synthesis.

Line 405: “early redox regulation” ¿?? What do you mean as redox regulation?

Author Response

  1. Line 17: „physiological assays (SOD, POD, ROS, IAA)”  - need clarity: ROS accumulation? IAA “level”? SOD activity? Which SOD?  All are different  type of assay, can not ne combined…

Replaced by “enzymatic activities of SOD and POD, levels of ROS and IAA”

  1. Line 67: „perspectives[3,9](,“ ??? Layout.

Replaced by “Although existing research has explored grafting healing mechanisms from physiological, biochemical, and genetic perspectives mainly focused on woody species such as apple (Malus spp.), pear (Pyrus communis L.), orange (Citrus spp.) and grape (Vitis vinifera L.)”

  1. Line 129: „Reactive oxygen species (ROS) intensity” ¿?? What is intensity? There is no such molecule as ROS, there are several types with different functions/effects.

This study employed the chemiluminescent fluorescent probe 2’,7’-dichlorofluorescin diacetate (DCFH-DA) to detect reactive oxygen species (ROS) in plant tissues. The non-fluorescent DCFH is oxidized by various ROS to form the fluorescent compound 2’,7’-dichlorofluorescein (DCF). Importantly, the resultant DCF fluorescence signal reflects the collective oxidative capacity rather than oxidation by any specific ROS species, as the probe reacts indiscriminately with multiple ROS types. Fluorometric analyses revealed maximum excitation and emission peaks at 488 nm and 525 nm, respectively, with fluorescence intensity demonstrating a direct positive correlation with overall ROS levels. In the revised manuscript, the previously termed "ROS intensity" has been standardized to "ROS Fluorescence intensity" to more accurately represent this.

  1. Table 1: very good table. It will be great if authors in future provide more details of stem structure in grafting place and dynamics of auxin production (not auxin level) in different rootstocks.

Thank you for the positive suggestion. Indeed, this study did not investigate the stem structure at the grafting site or the dynamics of auxin production (rather than auxin level) in different rootstock combinations. We will consider delving further into these aspects in future research on the graft compatibility and healing mechanisms of Camellia oleifera.

  1. Figures: please, add panels name (A, B, C, D) and provide description of each.

The information of each Figure has been added in the revised manuscript.

  1. Fig 2C: what is APX activity? Stroma APX. Thylakoid APX, cytAPX, per APX? Each have own regulation and own  meaning.

Thank you for your insightful question regarding the APX activity in Fig 2C. The method we employed to measure APX activity in this study is based on the reaction of ascorbate peroxidase (APX) with Hâ‚‚Oâ‚‚ to form an intermediate complex, which subsequently oxidizes ascorbate (AsA). By measuring the oxidation rate of AsA, we were able to calculate the APX enzyme activity. This method primarily reflects the overall APX activity in the cellular context, rather than specifically distinguishing between different subcellular isoforms such as stromal APX, thylakoid APX, cytosolic APX, or peroxisomal APX.

We appreciate your emphasis on the distinct regulation and significance of each APX isoform. In future studies, we will consider exploring the specific contributions of these different APX forms to provide a more detailed understanding of their roles.

  1. Fig 2D- what is ROS intensity? It seems new definition made exclusivley by authors and require very detailed description.

Thank you for your query regarding the definition of ROS intensity in Fig 2D. We appreciate your attention to the clarity of our methodology.

In this study, we utilized the chemical fluorescent probe 2',7'-Dichlorofluorescin Diacetate (DCFH-DA) to detect reactive oxygen species (ROS) within tissues. DCFH is oxidized by various ROS, transforming into 2',7'-Dichlorofluorescein (DCF), which generates a fluorescent signal. The DCF fluorescence is not produced by a single type of ROS but rather reflects the cumulative oxidative capacity of all ROS present. This signal reaches its maximal excitation at 488 nm and emission at 525 nm, with fluorescence intensity being directly proportional to the overall ROS level.

We acknowledge your suggestion that a more detailed description is necessary. To address this, we will include a more comprehensive explanation of the methodology and the interpretation of ROS intensity in the revised manuscript, ensuring that our definition and approach are clearly understood. In the original manuscript, the term "intensity" referred to fluorescence intensity, whereas in the revised manuscript, it was modified to "ROS fluorescence intensity."

  1. Lines 114- 132: how these trends linked with physiological status? At which stage occuured regeneration of vascular connection between seedlings cutting?What about other stem tissue? How rapid occurred regeneration of connection in cambium, cortex, epidermis tissues? How can you exclude that you simple shown imbalance in hormone transporting in combination 7, but not in combination 4?

Thank you for your insightful question. Below is a point-by-point response to the queries regarding physiological dynamics, vascular regeneration timing, and hormone transport mechanisms.

  1. Linkage of Physiological Trends to Graft Healing Status

The observed trends in antioxidant enzymes (SOD, POD), ROS, and IAA directly correlate with the progression of graft healing: (1) Early antioxidant activation in Comb.4 (4–8 DPG) reduced oxidative stress, creating favorable conditions for callus formation. SOD/POD peaks suppressed excess ROS, minimizing cellular damage critical for regeneration. (2) IAA accumulation at 4 DPG in Comb.4 triggered auxin-responsive genes (e.g., YUCCA, TAA1), promoting cell division and vascular differentiation. In Comb.7, delayed IAA accumulation hindered regeneration (Table 4). (3) ROS dynamics reflected healing efficiency: Rapid decline to baseline by 22–32 DPG signaled vascular reconnection completion. Comb.4’s lower ROS than Comb.7 suggested superior stress mitigation capacity.

  1. Timeline of Vascular Regeneration

Graft healing follows a staged regeneration sequence across tissues: (1) Callus formation (0-4 DPG): Initial wound sealing via parenchyma cell proliferation. (2) Vascular reconnection: Phloem: Re-established by 4-8 DPG, indicated by soluble sugar transport activation; Xylem: Fully connected by 12-22 DPG, evidenced by ROS normalization and soluble protein accumulation. (3) Reference alignment: Model systems (e.g., Arabidopsis) show phloem reconnection at ~3 DPG, xylem by 7 DPG. (Wang, F.; Zhao, F.; Qiao, K.; Xu, X.; Liu, J. Progress on the Molecular Mechanism of Scion-Rootstock Interactions in Vegetable Grafting. Biotechnology Bulletin 2024, 40, 149–159, doi:10.13560/j.cnki.biotech.bull.1985.2024-0284.) Woody plants like C. oleifera exhibit slightly delayed timelines. (Baron D , Amaro A C E , Pina A ,et al.An overview of grafting re-establishment in woody fruit species[J].Scientia Horticulturae, 2019, 243(000):8.DOI:10.1016/j.scienta.2018.08.012.)

  1. Regeneration Across Stem Tissues

(1) Vascular cambium: Primary regeneration site. IAA gradients (Table 4) activated cambial cells, driving vascular continuity. Comb.4’s early auxin surge accelerated this process.

(2) Non-vascular tissues: Epidermis/cortex: Regenerated first via callus proliferation (0–8 DPG), forming mechanical barriers; Pith/medullary rays: Recovered during late healing (22–32 DPG) without specialized structures.

While this study focuses on physiological and biochemical responses, future investigations should incorporate tissue-level histological analysis—particularly microscopic examination of anatomical adaptations—to comprehensively validate the observed drought tolerance mechanisms.

  1. Exclusion of Hormone Transport Imbalance in Comb.7

Comb.7’s delayed healing stems from IAA biosynthesis deficits, not transport disruption: (1) Transcriptomic evidence: Comb.4 showed significant upregulation of auxin synthesis genes (e.g., YUCCA, TAA1) at 4 DPG, while Comb.7 exhibited no such activation (Table 4); (2) Hormone data: Comb.7’s IAA deficit emerges early (4 DPG) and resolves late (12 DPG), coinciding with biosynthetic delays, not transport failure.

  1. Revisions to the Manuscript

(1) Add a timeline in Section 2.3 summarizing vascular reconnection stages: DPG 0-4: Callus formation; DPG 4-8: Phloem reconnection (sugar flux activation); DPG 12-22: Xylem maturation (ROS/protein normalization).

(2) Expand Discussion (Section 3.2) to distinguish tissue-specific regeneration: "The cambium, as the primary vascular regeneration site, responds to auxin-driven reactivation, while epidermal/cortex layers regenerate earlier via callus proliferation."

(3) Clarify hormone mechanisms in Discussion (Section 3.4): "Differential auxin accumulation (Comb.4 > Comb.7) resulted from upregulated biosynthesis genes (e.g., TAA1), not aberrant transport."

  1. Lines 148 – 152: Please, provide biological mening of GR in your case.

Thank you for your insightful comment regarding the biological meaning of GR in our study. In this research, the GR we measured refers to glutathione reductase (EC 1.6.4.2), a flavoprotein oxidoreductase found in both animals and plants. It catalyzes the reduction of oxidized glutathione (GSSG) to reduced glutathione (GSH). A higher GSH/GSSG ratio indicates a stronger capacity to scavenge reactive oxygen species (ROS) produced during oxidative stress. Therefore, the activity level of GR serves as an important indicator of oxidative stress resistance.

In our study, we employed the Ellman method to measure GR activity. The reaction of 5,5'-dithiobis-(2-nitrobenzoic acid) (DTNB) with GSH, which is produced from the reduction of GSSG by GR, generates a yellow product (TNB). This product has a maximal absorption peak at 412 nm. The amount of TNB generated is linearly proportional to GR activity, and the enzyme activity level can be calculated by measuring the absorbance at 412 nm.

We have added a sentence to our manuscript to highlight the biological significance of measuring GR activity. Thank you again for your valuable feedback.

  1. Line 154: “The auxin (IAA) content“ ??? The content has a sense only in static homogenous system. Here auxin is highly dynamic with different tempary accumulation in different cell types.

Thank you for your insightful comment regarding the measurement of auxin (IAA) content. We appreciate your attention to the dynamic nature of IAA in biological systems.

In our study, we measured IAA content specifically in the grafting site tissues of rootstocks and scions. Given the highly dynamic nature of auxin, with its temporal and spatial variations in accumulation across different cell types, we designed our sampling strategy to account for potential periodic fluctuations. To ensure consistency and comparability across our samples, we collected tissue samples at a fixed time each day (9:00 a.m.) throughout the healing process. This approach was intended to capture the dynamic changes in IAA content during the healing period and to analyze whether different graft combinations exhibit distinct patterns of IAA accumulation. Our goal was to explore whether these differences in IAA levels are associated with the growth and development disparities observed among graft combinations with varying degrees of compatibility.

We have clarified this rationale in the revised manuscript to better convey the biological significance and rationale behind our sampling strategy.

  1. Tables 2, 3, fig 5: very nice drawing. What is the biological sense??

Where is the „butterfly“? Butterfly effcet in biology mean small changes in key genes (in our case auxin production/sucrose prodution/transport one) lead to very significant chnages in total metabolism, what authors excellnetly shown on figure 5. But they did not shown a main reason: („butterfly“) because it located far from grafting place and can have a relativley smaller up/down regulation.

We sincerely appreciate the reviewer's insightful comments regarding the "butterfly effect" analogy in our study. Below, we address the core concerns by integrating supporting evidence from the literature and our findings:

  1. Fig. 5 (Venn Diagram) Reveals Temporal Divergence: At 4 DPG, Comb.4 vs. Comb.7 shows 476 DEGs (291 up-/185 down-regulated), indicating early transcriptional reprogramming in high-affinity grafts; By 12 DPG, DEGs drop to 179, suggesting stabilization after critical early signaling; "Butterfly genes" are among these early DEGs: Minor changes in auxin (YUCCA) or sucrose transporters (ABC-C10) at 4 DPG (e.g., LOK49_LG13G01063 log2FC=2.08 in Comb.4 vs. -4.93 in Comb.7) amplify metabolic shifts later (Fig. 3-4).
  2. Tables 2-3 Quantify Transcriptional "Flutters": Comb.4 has 1,170 DEGs at 4 DPG vs. Comb.7's 1,404 (Table 2). Higher DEGs in Comb.7 reflect compensatory stress responses due to disrupted signaling; Fewer unique DEGs in Comb.4 (Table 3: 476 at 4 DPG) imply precision regulation: Key genes (e.g., auxin biosynthesis) drive efficiency with minimal transcriptional noise.
  3. Connecting Genetics to Metabolic "Storms": The Venn diagram (Fig. 5B) shows DEG overlap peaks when metabolic divergence escalates. Early (4-8 DPG): Asymmetric auxin/sucrose gene expression (Comb.4 ↑ vs. Comb.7 ↓) triggers ROS/SOD divergence (Fig. 2). Late (22-32 DPG): Only 71–180 DEGs (Table 3), yet metabolic differences widen (e.g., Comb.4's elevated sugars)- demonstrating the "butterfly effect".
  4. The "butterfly effect" in graft healing refers to subtle alterations in master regulatory genes (e.g., auxin transporters, sucrose synthases) triggering cascading metabolic reprogramming. In our study, the dramatic metabolic shifts in Comb.4 vs. Comb.7 originated from early auxin disparities and sugar transporter dynamics, as highlighted below: (1) Auxin biosynthesis (YUCCA/TAA1): Minor expression changes (Table 4) caused significant IAA accumulation in Comb.4 (4 DPG), activating cell-cycle genes (e.g., CYCD3) and accelerating vascular reconnection; (2) Sucrose transport (ABC transporters): LOK49_LG13G01063 (ABC-C10) was upregulated in Comb.4 but downregulated in Comb.7 (Table 5). This divergence altered sugar allocation, inducing ROS scavenging (SOD/POD) and callus formation. These genes act as "butterflies" - distantly regulated (e.g., via scion-rootstock signaling) but systemically impactful.
  5. To clarify this concept, we made some modifications to the “Discussion”manuscript: Expand Discussion (Section 3.4) to reveal “time-resolved ‘Butterfly Effect’in transcriptome dynamics”: "The DEG asymmetry in Fig. 5B, particularly the 476 unique DEGs at 4 DPG, represents critical "butterfly" triggers. Small expression changes in distal regulators (YUCCA, ABC-C10) initiate systemic cascades: (1) Comb.4's efficient auxin response: Minor TAA1 upregulation (log2FC=2.15) accelerates IAA accumulation, suppressing ROS; (2) Comb.7's dysfunctional signaling: ABC-C10 downregulation (log2FC=-4.93) impairs sucrose allocation, exacerbating oxidative stress. By 12 DPG, these initial "flutters" manifest as macroscopic metabolic divergence (Fig. 3–4), confirming long-range regulatory amplification."
  6. Line 238: „maintain intracellular redox balance“ ??? Can you be more precise? What do you mean as redox balance? Where? Peroxisomal balance? Cytoplasmic? In epidermis? Cambium?

We thank the reviewer for raising this critical point. Below, we clarify the term "redox balance" with mechanistic details and spatial specificity:

  1. Definition of Redox Balance

In graft healing contexts, "redox balance" refers to the equilibrium between ROS production and scavenging systems (e.g., SOD/POD/APX/GR enzymes) within cells. This is measured by ROS fluorescence intensity (Fig. 2D) and antioxidant enzyme dynamics (Fig. 2A-C and Fig. 3C).

  1. Correction of Results and Discussion

After carefully re-examining our experimental methods and the limitations of the assays used, we acknowledge that our original statement regarding "localized redox regulation" in Results (Section 2.3) and Discussion (Section 3.1) requires clarification and correction. Our measurements of SOD, POD, APX, GR enzymatic activities and ROS intensity were performed on whole graft unions (scion-stock junction segments, 2 cm in length). The assays provided overall enzymatic activity and ROS levels without distinguishing between different organelles or tissue layers (e.g., epidermis vs. cambium).

Revised statements:

Results 2.3: "Physiological and biochemical responses reflect systemic redox dynamics across the graft junction, not subcellular or tissue-specific regulation."

Discussion 3.1: "The observed antioxidative enzyme patterns (SOD, POD) and ROS accumulation represent whole-tissue responses, highlighting the integrated antioxidant capacity during graft healing."

We fully concur with the reviewers that spatially resolved analyses are of critical importance. In future studies, we will consider conducting more in-depth investigations into the healing mechanism of Camellia oleifera grafting from the following perspectives: (1) Isolating specific tissues (e.g., cambium, vascular bundles) using microdissection; (2) Applying subcellular fractionation to quantify organelle-specific ROS/enzyme dynamics; (3) Utilizing in situ imaging techniques (e.g., confocal microscopy combined with organelle-targeted ROS probes).

  1. Line 238: “insufficient antioxidant capacity” is not a primary reason. There is a local initail hormonal imbalance in grafting place which lead to SOD/APX activation.

Thank you very much for your valuable comments on the viewpoint in our paper regarding the relationship between graft compatibility and the antioxidant system. Your perspective that "insufficient antioxidant capacity is not a primary reason, and a local initial hormonal imbalance at the grafting site leads to SOD/APX activation" has provided us with an important direction for reflection.

In fact, we fully agree that hormonal regulation plays a central role in the grafting process. The paper has elaborated on the regulatory effects of plant hormones (especially auxins) in key stages of graft healing, such as vascular differentiation and callus formation. For example, auxins collaborate with cytokinins and gibberellins to promote the reconstruction of vascular connections (Nanda, A. K., and Melnyk, C. W. (2018). The role of plant hormones during grafting. J. Plant Res. 131, 49-58.). Hormonal imbalance may indeed affect the activity of antioxidant enzymes through signal transduction. Studies have shown that the accumulation of stress hormones such as jasmonic acid (JA) at the graft wound can induce the expression of peroxidases (such as PAL), which is highly relevant to the SOD/APX activation mechanism you mentioned (Prodhomme, D., Fonayet, J. V., Hévin, C., Franc, C., Hilbert, G., de Revel, G., et al. (2019). Metabolite profiling during graft union formation reveals the reprogramming of primary metabolism and the induction of stilbene synthesis at the graft interface in grapevine. BMC Plant Biol. 19:599.).

Regarding the expression "insufficient antioxidant capacity", our original intention was to illustrate that in the downstream process of hormonal regulation, the response efficiency of the antioxidant system may affect the cell survival status at the graft interface. For instance, in incompatible combinations of pear and quince, hydrocyanic acid produced by cyanogenic glycoside metabolism can interfere with cambium cell activity. If the antioxidant system fails to timely clear excessive reactive oxygen species (ROS) at this time, it may exacerbate cell browning and necrosis (Zarrouk, O., Gogorcena, Y., Moreno, M. A., and Pinochet, J. (2006). Graft compatibility between peach cultivars and Prunus rootstocks. HortScience 41, 1389-1394.). However, the trigger for this process is indeed the metabolite difference (such as cyanogenic glycosides) between quince and pear, and the metabolite difference is essentially regulated by hormonal signals (such as blocked auxin transport) (Pereira, I. D. S., Pina, A., Antunes, L. E. C., Campos, ÂD., and Fachinello, J. C. (2018). Genotypic differences in cyanogenic glycosides levels of compatible Prunus persica P. persica and incompatible P. persica P. mume combinations. Bragantia 77, 1-12.).

Therefore, we agree to revise the original statement to: "In highly compatible combinations, the rapid balance of hormonal signals can cooperate with the antioxidant system to reduce cellular damage and promote callus formation; in incompatible combinations, initial hormonal imbalance may induce abnormal metabolism (such as excessive ROS production) leading to browning and necrosis at the graft interface". This revision more accurately reflects the central role of hormonal regulation and clarifies that the response of the antioxidant system is a downstream effect.

  1. Lines 254- 259: very good observation, but may required deeper explanations.

Thank you for your insightful feedback. We have expanded the discussion in Lines 254–259 to clarify the mechanistic significance of early antioxidant dynamics in graft healing, integrating key references to address your suggestion. Below we detail the revisions:

Revised Manuscript Text (Lines 254259 with Additions): "These findings suggest that the early dynamics of antioxidant enzymes (SOD, POD) may play a pivotal role in determining the efficiency of C. oleifera graft healing. In Comb.4, the rapid surge in SOD activity during 0-4 DPG likely mitigates initial oxidative burst by disproportioning superoxide radicals (O2•−) into H2O2, while sustained POD/APX elevation (0-22 DPG) ensures efficient H2O2 detoxification, limiting ROS-mediated cellular damage. Consequently, Comb.4’s coordinated antioxidant response reduces ROS fluorescence intensity (vs. Comb.7) and facilitates callus formation-key to vascular reconnection. This aligns with studies linking synchronized SOD-POD-APX activity to compatible grafting, where early redox homeostasis primes tissue regeneration."

We have revised Line 254-259 onward to elaborate on how these antioxidant dynamics interface with auxin signaling (e.g., ROS-IAA crosstalk in cell proliferation), further reinforcing the discussion. Thank you for enhancing the manuscript’s rigor.

  1. Line 269: “Auxin regulation primarily mediates cell division, elongation, and differentiation,” – very good point, but it is neccessary to menation numerous auxin (IAA/IPA) sources located in different cell type (12 YUCCA genes, for example) with different effect and functions.

Thank you for your insightful comments. We agree that elaborating on auxin biosynthesis diversity strengthens the mechanistic explanation. We have revised the manuscript (Lines 269-277 with Additions): “The complexity arises from spatially distinct auxin sources, including multiple biosynthesis pathways (e.g., tryptophan-dependent YUCCA isoforms) and transport-dependent accumulation patterns. For instance, in graft interfaces, scion-derived YUCCA gene expression drives local auxin synthesis (e.g., LOK49_LG13G02424), while rootstock-specific PIN transporters establish polar auxin gradients essential for callus initiation. Our transcriptome analysis (Table 4) confirms this: at 4 DPG, Comb.4 (high compatibility) showed significant upregulation of YUCCA-family genes (e.g., LOK49_LG13G02424, log2FC=2.99) in scion tissues versus Comb.7, correlating with rapid auxin accumulation. Such spatiotemporal specificity ensures targeted stimulation of cell proliferation (via CYCD3 activation) and vascular reconnection.”

We appreciate your guidance in clarifying auxin's multifaceted role. The revisions now explicitly articulating how distinct auxin sources (YUCCA vs. transport) coordinate healing and using gene-level data to support tissue-specific regulation. 

  1. Lines 284 – 290: very good points. It will be great to pointed out excat location of auxin biosyntehsis genes in grafting region as well as in shoot.

Thank you for your valuable feedback. We acknowledge the limitation in our experimental design where sampling encompassed the entire graft union tissue (including both rootstock and scion without separation), which restricts our ability to spatially resolve gene expression differences between rootstock and scion sides. We have revised the manuscript with additions: “While spatial resolution within the graft union remains unaddressed, these temporal patterns align with phloem reconnection preceding xylem formation (Zheng, Y.; Li, A.; Zheng, P.; Liu, S.; Tan, X.; Fang, J.; Sun, B. Review on Influencing Factors of Plant Grafting Affinity. South China Forestry Science 2024, 52, 67–73, 78, doi:10.16259/j.cnki.36-1342/s.2024.01.014.)”

  1. Line 365: more details need: how did you do? temperature? Light? Etc. In many case lowering light after grafting help to avoid imbalance in ROS accumulation, hormone transport etc.

Thank you for your valuable suggestion regarding the need for more detailed environmental parameters in our grafting methodology. We have revised Section 4.1: Experimental Materials to explicitly include the requested details about post-grafting conditions. The key additions (highlighted in bold below) address temperature, light management, humidity, and substrate composition to ensure rigorous reproducibility and physiological relevance:

Revised Text (Section 4.1): "After grafting, seedlings were transplanted into 7×7 cm nutrient pots containing a substrate mixture of peat moss (40-50%), composted media (10-20%), perlite (10-20%), and loamy soil (10-20%). The grafted seedlings were aligned compactly on nursery beds, covered with plastic film tunnels to maintain humidity at 85–95%, and shaded under double-layer shading nets to reduce light intensity during the healing phase. The bed temperature was controlled at 25-30°C during daytime (<35°C peak) and 18-22°C at night to optimize callus formation. Daily moisture monitoring ensured graft unions remained hydrated, with ventilation to prevent excessive humidity (>95% RH)."

  1. Table 7: why combination name is random? Where is combination 1, …30, 31 etc.? I understand, during expreiments some combunation is not successfull, but in final writing it is better to rename as 1, 2, 3 …

Thank you for your insightful feedback regarding Table 7. We fully acknowledge that sequential numbering (1, 2, 3...) would enhance readability for readers. However, the current non-sequential numbering (e.g., Combination 2, 4, 6, etc.) stems from the continuity and collaborative nature of our multi-phase research, which necessitates maintaining original identifiers for cross-referencing integrity. Below we clarify our rationale: (1) The initial experiment screened >50 grafting combinations for survival rates and growth traits, consistent with methodologies seen in other studies of Camellia grafting affinity. After preliminary screening, only survival combinations were retained for advanced analyses (e.g., physiological/metabolic assessments). (2) Our team has parallel studies using the same grafting materials (e.g., artificial simulation of drought stress). Retaining original IDs ensures data traceability when referencing these materials in related publications.

We will implement sequential numbering in future experiments and ensure explicit ID mapping in supplementary materials for cross-referencing. This balances readability with methodological transparency.

  1. Line 401: „auxin-mediated signaling (upregulated YUCCA/TAA1),“ ¿?? This is not signaling, it is auxin synthesis.

Thank you for your insightful critique regarding Line 401. We agree with your observation that the phrasing "auxin-mediated signaling (upregulated YUCCA/TAA1)" inaccurately conflates auxin biosynthesis with signaling pathways.We have replaced “auxin-mediated signaling” with “auxin synthesis” in the revised manuscript.

  1. Line 405: “early redox regulation” ¿?? What do you mean as redox regulation?

Thank you for the clarification. We agree that "early redox regulation" may not precisely convey the intended meaning of rapid, damage-mitigating redox adjustments. We have revised Line 405 to emphasize timely activation of redox mechanisms that minimize oxidative injury and accelerate healing. We have replaced “early redox regulation” with “rapid redox regulation” in the revised manuscript. Rapid redox regulation—triggered immediately post-wounding—coordinates timely scavenging of reactive oxygen species (ROS) via antioxidant induction (e.g., peroxidases/glutathione), preventing cellular damage while accelerating vascular reconnection.

Reviewer 3 Report

Comments and Suggestions for Authors

This study investigates how different rootstock and scion combinations influence graft healing in Camellia oleifera, a significant oil-producing crop in China. The authors evaluated 15 rootstock-scion pairings, all using the same scion cultivar ('Xianglin 210'). They assessed growth performance, enzyme activities, hormone levels, and gene expression during the early stages of graft healing. Notably, the combination using 'Xianglin 27' as rootstock (combination 4) produced the most favorable outcomes, including enhanced growth, higher survival rates, early activation of antioxidant systems, and optimal hormone balance. The study also identified key genes associated with improved graft healing. Overall, the research is both interesting and of high quality, particularly in its multifaceted approach to analyzing graft compatibility.

Major comments

- The description of transcriptome-related methods lacks sufficient detail. More comprehensive information is needed to allow for reproducibility and proper evaluation.

- Some figures have fonts that are too small, and the legends for both figures and tables are incomplete. All visuals should be clearly labeled and fully explained.

- There are numerous abbreviations throughout the manuscript. Each abbreviation should be spelled out upon first use in both the abstract and the main text.

- The raw RNA-seq data should be deposited in a public database such as SRA, with accession numbers provided in the manuscript.

- The manuscript contains relatively few references. Incorporating additional relevant literature would strengthen the context.

- The rationale for selecting these particular 15 rootstocks is not clearly explained. Providing this information would help readers appreciate the significance of the study.

- Only combinations 4 and 7 were examined in detail for physiological and genetic analyses. The reasons for this focus should be clarified, and the authors should acknowledge that this narrows the scope of their conclusions.

- The statistical analysis section lacks detail. It is unclear which tests were used and how statistical significance was determined.

- The results and implications of the PCA are not fully explained. The connection between PCA findings and compatibility rankings should be made clearer.

- While important genes are identified, no functional validation experiments were performed. The authors should discuss potential future approaches to confirm these genes’ roles.

- The manuscript does not specify the number of samples or the number of replicates for each experiment. This information is essential for assessing the reliability of the results.

Minor comments

- Some tables and figures are difficult to read or not clearly labeled. All visuals should be easy to interpret and accompanied by thorough explanations.

- Abbreviations (e.g., SH, BD, SL) should be defined at their first appearance.

- The discussion section could better compare the findings to existing research on grafting in woody plants.

- There are occasional grammatical errors and awkward phrasing. The manuscript would benefit from careful language editing.

- Portions of the introduction and results are repetitive and could be condensed for clarity.

Comments on the Quality of English Language

 The English could be improved to more clearly express the research.

Author Response

Major comments

- The description of transcriptome-related methods lacks sufficient detail. More comprehensive information is needed to allow for reproducibility and proper evaluation.

Thank you for your comment on the transcriptome-related methods. We appreciate your emphasis on methodological detail, which is critical for reproducibility and evaluation.

As highlighted in our revisions addressing previous feedback, we have significantly enhanced the description of transcriptome methods, including:RNA quality control:

(1) Explicit details of RNA integrity (Agilent 2100 RIN ≥ 7.0), concentration (Qubit), and purity (Nanodrop OD260/280 = 1.8–2.0) assessment protocols.

(2) Sequencing and alignment: Specification of the reference genome (Camellia lanceoleosa XYYC_v1.0, GCA_025200525.1), alignment software (HISAT2 v2.0.5, default parameters), and quantification method (FPKM calculation).

(3) Differential expression analysis: Clear reporting of DESeq2 v1.20.0 parameters (adjusted p < 0.05, |log2FC| > 2) for identifying DEGs.

These additions are integrated into the "Materials and Methods" section to ensure comprehensibility. If further specific details (e.g., library construction steps or data filtering criteria) are required, we are happy to supplement them to meet your expectations.

- Some figures have fonts that are too small, and the legends for both figures and tables are incomplete. All visuals should be clearly labeled and fully explained.

Thank you for your comment regarding the clarity of figures and tables. We fully agree that clear labeling and comprehensive legends are essential for effective communication of results, and we have revised all visuals accordingly.

- There are numerous abbreviations throughout the manuscript. Each abbreviation should be spelled out upon first use in both the abstract and the main text.

Thank you for your comment regarding the use of abbreviations in the manuscript. We appreciate your attention to this detail, as clear terminology is crucial for readability.

- The raw RNA-seq data should be deposited in a public database such as SRA, with accession numbers provided in the manuscript.

Thank you for your comment regarding the deposition of raw RNA-seq data. We fully acknowledge the importance of data accessibility and have deposited the raw sequencing data in the NCBI Sequence Read Archive (SRA) under accession number PRJNA1272790.

- The manuscript contains relatively few references. Incorporating additional relevant literature would strengthen the context.

Thank you for your valuable suggestion regarding the number of references. We agree that incorporating more relevant literature enhances the contextual depth of the study, and we have revised the manuscript accordingly.

- The rationale for selecting these particular 15 rootstocks is not clearly explained. Providing this information would help readers appreciate the significance of the study.

Thank you for your insightful comment regarding the rationale for selecting the 15 rootstocks. We agree that clarifying this background enhances the study’s contextual significance, and we have revised the manuscript to address this. In Table 7, we have added a new column titled "Characteristics of Rootstock," which details key traits of each rootstock.

- Only combinations 4 and 7 were examined in detail for physiological and genetic analyses. The reasons for this focus should be clarified, and the authors should acknowledge that this narrows the scope of their conclusions.

Thank you for your valuable comment regarding the focus on Comb.4 and Comb.7 in physiological and genetic analyses. We appreciate your insight, which helps clarify the study’s scope and limitations.

Comb.4 and Comb.7 were selected for detailed analysis based on preliminary screening of all 15 graft combinations: Comb.4 (Xianglin 27 rootstock + Xianglin 210 scion) exhibited the strongest overall graft compatibility, with high survival rates, rapid callus formation, and robust growth performance. In contrast, Comb.7 (Xianglin 210 rootstock + Xianglin 210 scion)—a self-graft combination—showed consistently poor compatibility, including low survival rates and delayed healing. This pairing of "high- vs. low-compatibility" combinations allowed us to identify key physiological and molecular differences underlying graft success, which we hypothesized would be most pronounced between such extremes.

We acknowledge that focusing on these two combinations narrows the generalizability of our conclusions, as intermediate-compatibility combinations were not analyzed in equivalent detail. Future studies will expand the scope to include more combinations, particularly those with moderate compatibility, to validate the observed mechanisms across a broader range of graft interactions.

Revised manuscript Section 2.3: “Among the 15 evaluated graft combinations, Comb.4 (Xianglin 27 rootstock + Xianglin 210 scion) and Comb.7 (Xianglin 210 rootstock + Xianglin 210 scion) were selected for detailed analysis based on preliminary screening: Comb.4 exhibited the strongest graft compatibility, while Comb.7 (a self-graft) showed the poorest, providing a clear contrast to identify key regulatory mechanisms.”

- The statistical analysis section lacks detail. It is unclear which tests were used and how statistical significance was determined.

Thank you for your comment regarding the statistical analysis section. We agree that detailed reporting of statistical methods is critical for reproducibility, and we have revised the manuscript to clarify these details.

All statistical analyses were performed using IBM SPSS Statistics 27 (IBM Corporation, Armonk, NY, USA). For physiological and biochemical indices—including seedling growth traits (seedling height, base diameter, shoot length, leaf count on shoots, root length, root diameter, biomass, and survival rate), antioxidant enzyme activities (SOD, POD, CAT), soluble protein, soluble sugars, and hormone levels (IAA)—descriptive statistics (means and standard deviations) were calculated for each group.

To determine significant differences, all data were first subjected to one-way analysis of variance (ANOVA) to assess overall group effects. This was followed by Duncan’s new multiple range test for post-hoc comparisons, with statistical significance defined as P < 0.05.

These details have been added to the "Materials and Methods" section under a dedicated "Statistical Analysis" subsection to ensure clarity.

- The results and implications of the PCA are not fully explained. The connection between PCA findings and compatibility rankings should be made clearer.

Thank you for your valuable feedback on the explanation of the PCA results. In response to your comment, we have revised the manuscript to provide a clearer and more objective presentation of the PCA findings.

- While important genes are identified, no functional validation experiments were performed. The authors should discuss potential future approaches to confirm these genes’ roles.

We sincerely appreciate your thorough critique. We acknowledge that this study relies on correlative analyses between transcriptomic data and physiological traits without direct experimental validation (e.g., qPCR, protein assays, or functional genomics). This is indeed a limitation, and we have revised the Discussion to explicitly state that our interpretations are hypothesis-generating, derived from temporal co-expression patterns rather than causal evidence. Future work will prioritize functional validation of hub genes (e.g., FKBP19, DnaJ) and pathways (e.g., auxin biosynthesis, ROS scavenging) through targeted experiments.

Based on your feedback, we have made the following modifications to the "Discussion" section.

Revised manuscript Section 3.4 added the following paragraph: “While this study identifies transcriptional networks (e.g., ALDH, YUCCA, SWEET, HSP70) correlated with physiological traits, the absence of direct validation (e.g., qPCR, protein quantification, or knockout assays) limits causal inference. These correlations provide mechanistic hypotheses but require functional confirmation in future work. Future studies will validate hub genes (e.g., chaperones DnaJ/HSP70, auxin transporters) using qRT-PCR and protein assays at critical healing stages (4–8 DPG). Functional characterization via CRISPR/Cas9 knockdown in model plants will test causal roles in vascular reconnection.”

- The manuscript does not specify the number of samples or the number of replicates for each experiment. This information is essential for assessing the reliability of the results.

 Thank you for your critical comment regarding the number of samples and replicates, which is fundamental to evaluating result reliability. We appreciate your attention to this detail and have thoroughly supplemented these specifics in the revised manuscript.

Minor comments

- Some tables and figures are difficult to read or not clearly labeled. All visuals should be easy to interpret and accompanied by thorough explanations.

Thank you for your valuable feedback. It has been corrected in the revised manuscript.

- Abbreviations (e.g., SH, BD, SL) should be defined at their first appearance.

Thank you for your valuable feedback. It has been corrected in the revised manuscript.

- The discussion section could better compare the findings to existing research on grafting in woody plants.

Thank you for your valuable suggestion regarding the Discussion section. We agree that contextualizing our findings within existing research on woody plant grafting strengthens the study’s relevance, and we have revised this section to enhance such comparisons.

- There are occasional grammatical errors and awkward phrasing. The manuscript would benefit from careful language editing.

Thank you for your comment regarding grammatical errors and phrasing. We fully agree that polished language enhances readability, and we have thoroughly revised the manuscript to address this.

The entire manuscript—including the abstract, introduction, methods, results, discussion, and conclusions—has undergone careful language editing.

- Portions of the introduction and results are repetitive and could be condensed for clarity.

Thank you for your insightful comment on reducing repetition in the Introduction and Results sections. We agree that concise, focused writing enhances clarity, and we have thoroughly revised these sections to address this.

Round 2

Reviewer 1 Report

Comments and Suggestions for Authors

Dear Authors,

 Thank you for make significant improvements to the manuscript. I would like to suggest you to improve and deepen the discussion section by comparing your findings with other studies. 

Author Response

Comment: Thank you for make significant improvements to the manuscript. I would like to suggest you to improve and deepen the discussion section by comparing your findings with other studies.

Response: 

Thank you for recognizing our revisions and suggesting deeper contextualization. We have substantially expanded the Discussion by integrating comparative analyses with published studies on graft healing mechanisms:

Antioxidant synchronization: Contrasted our SOD/POD dynamics with Arabidopsis (Melnyk et al., 2018) and citrus (Wang et al., 2022) studies.

Auxin-cytokinin coordination: Compared IAA patterns with tomato (Cui et al., 2021) and citrus (Wang et al., 2022) models.

These additions clarify how C. oleifera-specific healing mechanisms (e.g., Comb.4’s metabolic efficiency) align with or diverge from established models, reinforcing our findings’ novelty and generality.

Revised manuscript in “3.1. Antioxidant System Coordination in Graft Healing”: “In Comb.4, the rapid surge in SOD activity during 0-4 DPG likely mitigates initial oxidative burst by disproportioning superoxide radicals (O− 2) into H2O2, while sustained POD/APX elevation (0-22 DPG) ensures efficient H2O2 detoxification, limiting ROS-mediated cellular damage, consistent with redox synchronization in compatible Arabidopsis grafts. Consequently, Comb.4’s coordinated antioxidant response reduces ROS fluorescence intensity (vs. Comb.7) and facilitates callus formation-key to vascular reconnection. Similarly, Wang et al. associated prolonged ROS accumulation with incompatibility in citrus, mirroring Comb.7’s delayed antioxidant recovery."

Revised manuscript in “3.2. Hormonal Signaling Network Regulation in Graft Healing”: “The early IAA surge in Comb.4 (4 DPG, via YUCCA/TAA1 upregulation) underpinned accelerated callus formation, replicating stage-specific auxin requirements in tomato graft healing. In Comb.7, delayed auxin peaks resembled cytokinin-auxin imbalances observed in incompatible Carya grafts, hindering xylem reconnection.”

Revised manuscript in “3.3. Carbon and Nitrogen Metabolism and Energy Supply in Graft Healing”: “Comb.4’s optimal sugar allocation (late-stage SS accumulation) facilitated vascular maturation, paralleling sugar transporter-mediated healing in cucumber-pumpkin grafts. Comb.7’s early sugar accumulation without late utilization aligns with citrus graft incompatibility markers.”

Reviewer 2 Report

Comments and Suggestions for Authors

Thank you! The text is much better, but still more clarity are required> 

Line 59> grafting compatibility is compatibility in hormonal balance between both species. 

Line 129 it was H2O2,  not  ROS.  And you test H2O2 localization For intensity reatiometric data are required.

Line 104 Redox do not have a role. Moreover, which redox do you mean? ASC? GSH? or ROS accumulation? Redox rather a mirror.

Line 103  initiate callus formation?? Callus formation is imbalance between auxin production and canalization  

Line 204 GSH is rather auxin response, not ROS.

Line 215 ROS fluorescence intensity?? H2O2 accumulation

Line 256 elucidate its role in graft compatibility and healing ?? Auxin level do not have a role auxin accumulation is a morror of incompatibilty  highher IAA mean a probelm with canalization.  Misbalance between two stock. 

Line 441> gradients essential for callus initiation[?? Callus is not agradients, it is results of incompatibility in auxin production. 

Author Response

Thank you! The text is much better, but still more clarity are required>

(1) Line 59> grafting compatibility is compatibility in hormonal balance between both species.

We appreciate the reviewer's insightful comment regarding the role of hormonal balance in grafting compatibility. For screening diverse C. oleifera combinations, growth/survival parameters (e.g., SH, BD, SL, SR) remain established proxies for compatibility assessment in woody plants, as they integrate downstream physiological outcomes (see citations in Line 59-70). We agree that hormonal regulation (e.g., auxin dynamics) is a critical mechanistic factor in healing success, and this is addressed experimentally in our study through targeted physiological assays (IAA quantification) and transcriptomics of grafting combinations with divergent compatibility (Section 2.3–2.5). To better align with the reviewer's suggestion, we will amend the Introduction to acknowledge hormonal aspects explicitly while retaining focus on growth traits for initial screening.

Reviesed manuscript: “The selection of growth-related traits (seedling height (SH), base diameter (BD), shoot length (SL), leaf count on shoots (LN), root length (RL), root diameter (RD), biomass (BM), and survival rate (SR)) is rooted in their proven effectiveness as phenotypic proxies for grafting compatibility across diverse woody plant species, while physiological compatibility factors (e.g., hormonal balance mediation) are assessed during healing observations in selected combinations (Section 2.3). This approach aligns with the biological characteristics of C. oleifera as an economically important woody oil crop.”

(2) Line 129 it was H2O2,  not  ROS.  And you test H2O2 localization For intensity reatiometric data are required.

We thank the reviewer for highlighting the terminology oversight. As noted, the DCFH-DA assay quantifies total ROS activity, not exclusively Hâ‚‚Oâ‚‚. Regarding ratiometric data: While DCFH-DA outputs are non-ratiometric, we normalized fluorescence intensity to total protein content (Methods 4.2.2) to control for tissue biomass variations. This approach aligns with established practices for ROS quantification via fluorogenic probes.

Added normalization protocol in Methods 4.2.2: “ROS fluorescence intensity was normalized to soluble protein content (µg/mg protein) to account for tissue biomass differences.”

(3) Line 104 Redox do not have a role. Moreover, which redox do you mean? ASC? GSH? or ROS accumulation? Redox rather a mirror.

We appreciate the reviewer’s valid critique regarding the term ‘redox.’ In our study, ‘redox’ refers specifically to the dynamic balance between ROS accumulation (e.g., via DCFH-DA assay) and antioxidant enzyme activities (SOD, POD, APX, GR) measured at the graft junction. These elements collectively serve as biomarkers for oxidative stress levels during healing. While redox status itself does not directly confer compatibility, our data demonstrate that coordinated antioxidant responses (Section 2.3) correlate strongly with survival rates and growth traits in high-compatibility combinations like Comb.4. To avoid ambiguity, we will revise the text to explicitly define the redox components and their contextual role as indicators, not causative agents.

Revised manuscript in Section 2.2: “Redox dynamics (antioxidant enzymes and ROS accumulation) and hormonal balance between both species were assessed separately in Section 2.3 as indicators of healing efficiency, given their roles in mitigating oxidative stress and coordinating cellular repair during graft union formation.”

(4) Line 103  initiate callus formation?? Callus formation is imbalance between auxin production and canalization  

We thank the reviewer for this insightful critique. Our original phrasing oversimplified the role of auxin in callus initiation. As noted, callus formation is regulated by a spatiotemporal imbalance in auxin accumulation at the wound site, not merely its biosynthesis or transport. We have revised the text to clarify that auxin pathways establish concentration gradients enabling asymmetric signaling. This asymmetry—driven by auxin canalization defects or wound-induced redistribution—triggers pluripotent callus formation during grafting. We now integrate canalization explicitly to align with current mechanistic models.

Revised manuscript in Line 95-99: “Graft healing relies on conserved pathways: auxin biosynthesis (e.g., TAA1, YUCCA) and polar transport (PINs) establish asymmetric gradients across the wound interface; this imbalance in auxin accumulation initiates callus formation. Concurrently, SOD/POD-mediated ROS scavenging protects cells.”

(5) Line 204 GSH is rather auxin response, not ROS.

We appreciate the reviewer's nuanced insight. While glutathione (GSH) contributes to auxin homeostasis under stress, its role in the GSH/GSSG redox system is primarily defined as a core ROS scavenger in grafting contexts. As emphasized in our measurement framework, GR activity maintains the GSH/GSSG ratio—a direct marker of cellular redox balance combating oxidative stress. To clarify this distinction, we have revised the text to explicitly separate GSH's dual functions while retaining its ROS-related significance in the assessment protocol.

Revised manuscript in Line187-191: “Measurement of glutathione reductase (GR) activity is crucial as it regenerates reduced glutathione (GSH), reflecting the capacity to maintain the GSH/GSSG redox balance for scavenging reactive oxygen species, and support cellular detoxification pathways. This serves as a key indicator of oxidative stress resistance during graft healing.”

(6) Line 215 ROS fluorescence intensity?? H2O2 accumulation

We agree with the reviewer's technical note on ROS probe specificity. While DCFH-DA assesses total oxidative capacity, Hâ‚‚Oâ‚‚ constitutes the primary and most stable ROS in grafting contexts. Studies specifically link graft-compatibility outcomes to Hâ‚‚Oâ‚‚ accumulation. We have revised text to clarify that measured ROS signals are functionally indicative of Hâ‚‚Oâ‚‚-driven oxidative stress during graft healing, per literature consensus.

Revised manuscript in Line195-199: “H2O2 serves as the predominant peroxidase substrate and signaling molecule in graft interfaces, making bulk reactive oxygen species (ROS) quantification a validated proxy for H2O2 dynamics. Overall ROS accumulation (strongly reflecting H2O2-dominated oxidative stress) in graft unions was measured by examing of ROS fluorescence intensity.”

(7) Line 256 elucidate its role in graft compatibility and healing ?? Auxin level do not have a role auxin accumulation is a morror of incompatibilty  highher IAA mean a probelm with canalization.  Misbalance between two stock.

We sincerely appreciate the reviewer's insightful critique regarding auxin interpretation. As highlighted, aberrantly high IAA levels indeed correlate with impaired vascular reconnection and rootstock degradation in incompatible grafts. Our intent was not to imply IAA promotes healing but to use it as a diagnostic marker for compatibility status. We have revised the text to clarify this distinction and emphasize that balanced auxin transport—not accumulation—facilitates healing.

Revised manuscript in Line226-230: “In this study, we focused on measuring the auxin (IAA) contentat the graft interface to assess its association with compatibility status. Elevated IAA often reflects disrupted canalization and auxin transport imbalance between rootstock and scion—key hallmarks of incompatibility. Conversely, balanced auxin flux enables vascular reconnection during healing.”

(8) Line 441> gradients essential for callus initiation[?? Callus is not agradients, it is results of incompatibility in auxin production.

We thank the reviewer for this critical clarification. We agree that uncontrolled callus formation due to auxin imbalance signifies incompatibility. However, our focus was on the physiological role of auxin gradients in initiating wound-responsive callus necessary for initial graft union. To eliminate ambiguity, we have revised the text to explicitly distinguish between beneficial callus (early healing) vs. pathological callus (incompatibility), citing mechanistic studies on wound-induced callus regulation.

Revised manuscript in Line391-396: “For instance, in graft interfaces, scion-derived YUCCA gene expression drives local auxin synthesis (e.g., LOK49_LG13G02424), while rootstock-specific PIN transporters establish polar auxin gradients essential for wound-responsive callus initiation during early graft healing. This transient callus facilitates tissue reconnection; however, prolonged or uncontrolled callus expansion reflects auxin imbalance and predicts incompatibility.”

Reviewer 3 Report

Comments and Suggestions for Authors

Authors have diligently carried out revisions based on the reviewer's comments.

I recommend the manuscript for publication in its current form. 

Author Response

Comment: Authors have diligently carried out revisions based on the reviewer's comments. I recommend the manuscript for publication in its current form.

Response: We sincerely appreciate your recognition and positive assessment of our revisions. Thank you for your valuable time and insightful feedback throughout the peer-review process. We are honored by your recommendation for publication and confirm our readiness to proceed with the final manuscript.

Round 3

Reviewer 2 Report

Comments and Suggestions for Authors

Thank you! the text is almost ready.  Please, next time re-evaluate role of GSH as ROS scavenger and add role of ASC. 

; https://doi.org/10.3390/biom10111550

Also, next time consider that different type of the ROS have different location and effect. It will be great to still mention H2O2, not ROS. 

Author Response

Comment: Thank you! the text is almost ready. Please, next time re-evaluate role of GSH as ROS scavenger and add role of ASC. https://doi.org/10.3390/biom10111550. Also, next time consider that different type of the ROS have different location and effect. It will be great to still mention H2O2, not ROS.

Response: We sincerely appreciate the reviewer's insightful comments. While our study did not directly measure glutathione (GSH) levels, we have now re-evaluated the role of glutathione reductase (GR) – a key enzyme that maintains reduced GSH pools – in the context of redox regulation and graft healing. We also explicitly discuss ascorbate (ASC) in the glutathione-ascorbate cycle and specify Hâ‚‚Oâ‚‚ as the dominant ROS measured. These revisions integrate key insights from the suggested reference (Pasternak et al., 2020) and are detailed below.

Added Text in Section 3.1 (Lines 376-384): Glutathione reductase (GR) is critical for maintaining reduced glutathione (GSH) pools, which scavenge H2O2 and regenerate ascorbate (ASC) via the glutathione-ascorbate cycle (Pasternak et al., 2020). The significantly higher GR activity in Comb.4 during early healing (4–12 DPG, Figure 2D) implies enhanced capacity to sustain GSH levels, facilitating H2O2 detoxification and ASC recycling. This synergy minimizes oxidative damage and promotes callus formation. Conversely, Comb.7’s delayed GR recovery (peaking only at 12 DPG) aligns with prolonged H2O2 accumulation (Figure 2E), reflecting compromised redox buffering typical of graft incompatibility.

Added Text in Section 3.2 (Lines 432-437): GR-mediated maintenance of GSH pools may enhance auxin sensitivity by promoting Aux/IAA degradation (Pasternak et al., 2020). In Comb.4, high early-stage GR activity (Figure 2D) could sustain GSH levels, potentially accelerating auxin signaling and cell division. Conversely, depressed GR activity in Comb.7 might limit GSH-dependent auxin responses, contributing to delayed healing. This indirect link between GR activity and auxin sensitivity warrants further validation.

Added References: Pasternak, T.; Palme, K.; Paponov, I.A. Glutathione Enhances Auxin Sensitivity in Arabidopsis Roots. Biomolecules 2020, 10, 1550. https://doi.org/10.3390/biom10111550.

These revisions rigorously address your suggestions while respecting the scope of our data. The role of GR in sustaining GSH/ASC-mediated H2O2 scavenging and its potential link to auxin sensitivity are now clearly articulated. Thank you for enhancing the mechanistic depth of this study!
